# Galectin-3 impairs calcium transients and β-cell function

Qian Jiang[1,2,3,10], Qijin Zhao [1,2,3,10], Yibing Chen[1,2,3,10], Chunxiao Ma[1,2,3,10], Xiaohong Peng[4,5], Xi Wu[6,7], Xingfeng Liu [1,2,3], Ruoran Wang[8], Shaocong Hou [1,2,3], Lijuan Kong [1,2,3], Yanjun Wan[1,2,3], Shusen Wang [9], Zhuo-Xian Meng [8], Bing Cui [1,2,3], Liangyi Chen [4,5,7] & Pingping Li [1,2,3] ✉

In diabetes, macrophages and inflammation are increased in the islets, along with β-cell dysfunction. Here, we demonstrate that galectin-3 (Gal3), mainly produced and secreted by macrophages, is elevated in islets from both high-fat diet (HFD)-fed and diabetic *db/db* mice. Gal3 acutely reduces glucose-stimulated insulin secretion (GSIS) in β-cell lines and primary islets in mice and humans. Importantly, Gal3 binds to calcium voltage-gated channel auxiliary subunit gamma 1 (CACNG1) and inhibits calcium influx via the cytomembrane and subsequent GSIS. β-Cell CACNG1 deficiency phenocopies Gal3 treatment. Inhibition of Gal3 through either genetic or pharmacologic loss of function improves GSIS and glucose homeostasis in both HFD-fed and *db/db* mice. All animal findings are applicable to male mice. Here we show a role of Gal3 in pancreatic β-cell dysfunction, and Gal3 could be a therapeutic target for the treatment of type 2 diabetes.

The rates of obesity-associated type 2 diabetes mellitus (T2DM) have dramatically increased worldwide[1]. The pathogenesis of T2DM involves both insulin resistance and pancreatic β-cell dysfunction[2]. Generally, in the prediabetic, insulin-resistant state, compensatory increases in pancreatic β-cell mass result in an increase in insulin secretion to maintain normal glucose levels. However, with time, the β cells begin to fail, and insulin secretion cannot offset insulin resistance, leading to augmented plasma glucose levels and diabetes. People with maximum or near maximum insulin resistance have lost more than 80% of their β-cell function[3]. Therefore, progressive β-cell failure determines the progression of diabetes[3,4]. This β-cell dysfunction is mainly characterized by impaired glucose-stimulated insulin secretion (GSIS). Whenever insulin secretion is discussed, it should be noted that there are 2 phases, that is, the first phase and the second phase[4]. It has

been reported that β-cell dysfunction occurs in the earliest stages of T2DM, and loss of GSIS occurs in the first phase[4].

Obesity is a chronic low-grade inflammatory condition[5]. In addition to adipose tissue, liver and muscle inflammation[6,7], islet inflammation has been observed in many mouse models of obesity and T2DM as well as in human islets from patients with obesity and T2DM[5,8]. A study on chronic inflammation induced by obesity found that the chronic inflammation of islets caused by obesity is mainly controlled by macrophages in two obesity mouse models, with limited participation of adaptive immune cells (such as T cells and B cells)[9] or neutrophils and lymphocytes[10–12]. Macrophages show significant functional heterogeneity; M1-like macrophages have a proinflammatory effect, whereas M2-like macrophages have an anti-inflammatory effect. It has been reported that M1-like macrophages

[1]State Key Laboratory of Bioactive Substance and Function of Natural Medicines, Institute of Materia Medica, Chinese Academy of Medical Sciences and Peking Union Medical College, Beijing 100050, China. [2]Diabetes Research Center of Chinese Academy of Medical Sciences, Beijing 100050, China. [3]CAMS Key Laboratory of Molecular Mechanism and Target Discovery of Metabolic Disorder and Tumorigenesis, Beijing 100050, China. [4]College of Future Technology, Institute of Molecular Medicine, National Biomedical Imaging Center, Peking University, Beijing 100871, China. [5]Beijing Key Laboratory of Cardiometabolic Molecular Medicine, Peking University, Beijing 100871, China. [6]State Key Laboratory of Membrane Biology, College of Future Technology, Institute of Molecular Medicine, Peking University, Beijing 100871, China. [7]Peking-Tsinghua Center for Life Sciences, Peking University, Beijing 100871, China. [8]School of Basic Medical Sciences, Zhejiang University, Hangzhou, Zhejiang, China. [9]Organ Transplant Center, Tianjin First Central Hospital, Nankai University, Tianjin 300192, China. [10]These authors contributed equally: Qian Jiang, Qijin Zhao, Yibing Chen, Chunxiao Ma. ✉e-mail: lipp@imm.ac.cn

accumulate in the pancreatic islets of obese (*db/db*, *ob/ob*, or high-fat diet (HFD)-fed) mice. For example, in HFD-fed mouse islets, M1-like macrophages account for approximately 68.3% of the islet immune compartment[9,13]. These macrophages play a central role in the chronic inflammation of islets, and insulitis induces β-cell dysfunction. The proinflammatory factors produced by M1-like macrophages, such as TNFα and IL-1β, are increased in islets in obesity and T2DM, and they decrease GSIS by inducing NF-κB or JNK activation in β cells after 16 or 24 h of treatment[8,14,15]. Another mechanism by which macrophages decrease GSIS is direct cell-cell contact. Mouse intra-islet macrophages can engulf insulin secretory vesicles, and the number of engulfed vesicles is significantly increased in obesity[9]. Whether other factors and mechanisms from macrophages are involved in β-cell dysfunction (especially the acute mechanism) is still unknown.

Galectin-3 (Gal3) is a 30 kD protein, that consists of three structurally distinct domains containing a highly conserved short N-terminal domain (ND), a long ND and a C-terminal carbohydrate-recognition domain (CRD)[16]. The CRD forms a globular structure and accommodates a pocket for carbohydrate binding[16]. Gal3 is abundantly expressed and secreted by monocytes/macrophages and foam cells[17]. Our previous study found that Gal3 directly enhances

macrophage chemotaxis and induces insulin resistance in hepatocytes, adipocytes and myocytes[7]. The expression of Gal3 has been evaluated in M1-like macrophages in obesity[18] and in diabetic islets[13]. Moreover, Gal3 was found to trigger β-cell apoptosis and islet inflammation in T2DM mice[19,20]. Whether Gal3 directly and acutely inhibits β-cell GSIS has not been investigated.

In this study, we discovered that the Gal3/CACNG1 system functions as a regulatory pathway in the modulation of β-cell insulin secretion. The Gal3 concentration is increased in islets of mice/individuals with obesity and T2DM, and Gal3 acutely and directly decreases β-cell GSIS in vitro. This effect occurs through blocking calcium transients in β cells. Moreover, knockout of Gal3 improves islet morphology and glucose intolerance.

## Results
### Obesity increases Gal3 levels in islets and plasma
Consistent with previous findings that obesity increases plasma Gal3[7], circulating Gal3 levels were 3-fold elevated in HFD-fed obese mice (Fig. 1a). Next, we measured Gal3 levels in diabetic *db/db* (leptin receptor knockout) mice, which showed a decrease in β-cell function and rapid development of T2DM[21]. We found that plasma Gal3 levels

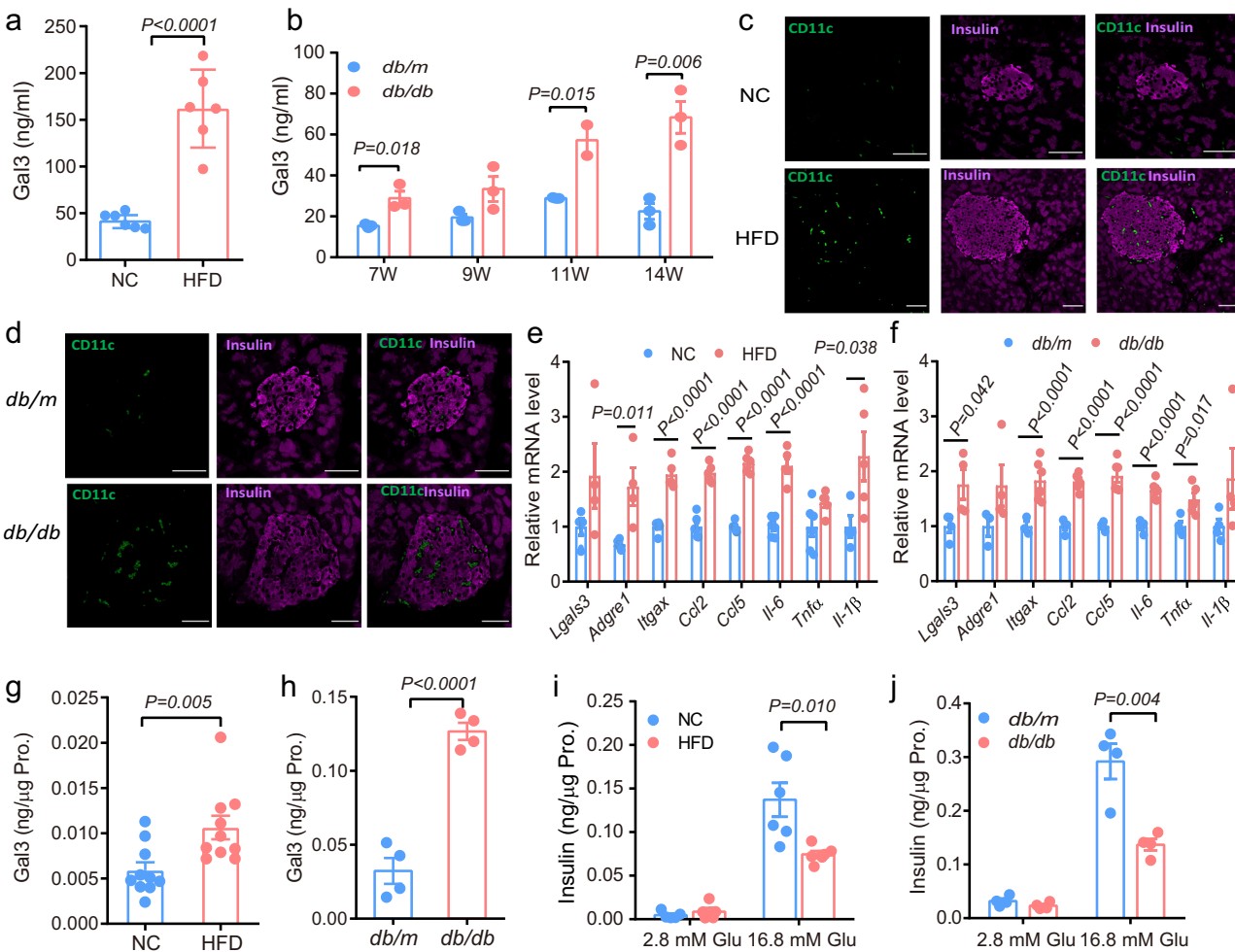

**Fig. 1 | Obesity increases Gal3 levels in islets and plasma. a** Gal3 levels in plasma of NC and HFD-fed mice (*n* = 6 mice). **b** Gal3 levels in plasma of *db/m* and *db/db* mice (*n* = 3 mice, 7 to 14 weeks of age). **c, d** Confocal image of CD11c by immunostaining in insulin-positive cells in pancreatic sections from NC and HFD-fed mice (**c**) or from *db/m* and *db/db* mice (**d**). *n* = 6 islets in 3 pancreatic sections, Scale bar, 50 μm. **e, f** Inflammatory gene expression in pancreas from HFD-fed (**e**) and *db/db* (**f**) mice. *n* = 4 mice for *Lgals3* and *Adgre1* and *n* = 5 mice in **e** for other genes. *n* = 4 mice for *db/m* and *n* = 6 mice in (**f**) for *db/db*. **g, h** Gal3 levels in primary islets from

HFD-fed (**g**) and *db/db* (**h**) mice. *n* = 10 mice (**g**), *n* = 4 mice (**h**). **i** GSIS in primary islets from NC and HFD-fed mice (n = 6 biologically independent samples). **j** GSIS in primary islets from *db/m* and *db/db* mice (n = 4 biologically independent samples). NC mice 20 weeks of age, HFD: 12 weeks from 8 weeks of age; *db/m* and *db/db* 28 weeks of age (**a, c–j**). Data were analyzed by two-sided Student's *t*-test without adjustments for multiple comparisons. All data are presented as the mean ± SEM. Source data are provided as Source Data file. NC Normal chow.

 

significantly increased with age in *db/db* mice (1.67-fold increase at 7 weeks old and 3-fold increase at 14 weeks old, Fig. 1b). Immunofluorescence staining of mouse pancreatic sections showed that pancreatic islets from HFD-fed and *db/db* mice were compensatorily enlarged with a looser and disorderly arrangement of β cells and α cells and had much more CD11c expression than that of islets from control mice (Fig. 1c, d and Supplementary Fig. 1a, b). Moreover, the expression of M1-like macrophage marker genes, such as *Adgre1* and *Itgax (CD11c)*, and other inflammatory factors, such as *Ccl2*, *Ccl5* and *Il6*, was significantly increased in the pancreases of HFD-fed and *db/db* mice (Fig. 1e, f). These results suggest that proinflammatory M1-like macrophages invade pancreatic islets in obesity and diabetes.

Given the increased M1-like macrophage accumulation in the pancreas during obesity and diabetes, we hypothesized that Gal3 levels would also be elevated in islets from obese or diabetic mice. In HFD-fed mouse islets, the Gal3 levels were significantly higher than those in normal chow (NC)-fed mice (Fig. 1g). The difference in Gal3 levels was more obvious between *db/db* and *db/m* mouse islets (Fig. 1h). Consistent with the differences in Gal3 levels, the gene expression of *Lgals3* was also increased in the pancreases of HFD-fed and *db/db* mice (Fig. 1e, f). Importantly, GSIS in islets isolated from either HFD-fed or *db/db* mice was 50% lower than that in control islets (Fig. 1i, j). In vivo, based on intraperitoneal glucose tolerance tests (IPGTTs) and insulin secretion tests, HFD-fed and *db/db* mice were characterized by impaired glucose tolerance and decreased first-phase insulin secretion (Supplementary Fig. 1c–f). Taken together, these results show that obesity and diabetes are accompanied by higher Gal3 levels in the pancreas, along with impaired β-cell function.

## Gal3 reduces islet and β-cell GSIS

Next, we asked whether Gal3 directly affected GSIS in β cells. To answer this question, we used different β-cell lines and islets isolated from normal and diabetic mice, followed by Gal3 treatment and GSIS-related experiments. The results showed that Gal3 treatment (80 or 250 ng/ml, similar to the levels found in HFD-fed mice) for 6 h markedly decreased the insulin secretion stimulated by high glucose (16.8 mM) in both rat INS-1 and mouse MIN6 cells, with no effect on intracellular insulin levels or cell viability (Fig. 2a, b and Supplementary Fig. 2a, b). Similar results were observed in islets from both C57BL6/J mice fed NC and *db/db* mice (Fig. 2c, d). Moreover, a lower Gal3 concentration (40 ng/ml) or shorter Gal3 treatment (1 h) still inhibited GSIS in MIN6 cells (Supplementary Fig. 2c, d). This inhibitory effect was further confirmed by islet perfusion experiments (Fig. 2e). It seems that Gal3 needs to be present all the time to induce the effect, since Gal3 pretreatment (1 or 6 h pretreatment, followed by 2 h of washing before the high-glucose stimulation) did not impair GSIS (Supplementary Fig. 2e). These results suggest that pathological levels of Gal3 can directly and acutely inhibit GSIS with no effect on the intracellular insulin content. Interestingly, in the process of GSIS in islets from NC mice, Gal3 inhibited glucagon and somatostatin secretion under low- and high-glucose conditions, respectively (Supplementary Fig. 2f, g). Gal3 also reduced insulin secretion in response to KCl, arginine and GLP-1 in mouse islets (Fig. 2f, g and Supplementary Fig. 2h).

Gal3 is mainly produced and secreted by monocytes/macrophages[17]. Consistent with this, we found colocalization of Gal3 and CD11c, an M1-like macrophage marker, in islets (Fig. 2h). This phenomenon was more obvious in *db/db* mice (Fig. 2i). In addition to Gal3, macrophages produce many inflammatory factors, such as IL-1β and TNFα, which are reported to inhibit GSIS after 24 or 48 h of treatment[22,23]. To test the contribution of Gal3 from macrophages to the inhibition of β-cell GSIS, we isolated intraperitoneal macrophages (IP-Macs) from wild-type (WT) and Gal3[−/−] mice and cocultured them with MIN6 cells (Fig. 2j), followed by GSIS tests. As shown in Fig. 2j and Supplementary Fig. 2i, j, MIN6 cells were surrounded by macrophages, similar to the finding in islets from HFD-fed and *db/db* mice. Both Gal3 levels and *Lgals3* gene expression were decreased in MIN6 cells and the

MIN6-Macs-knockout (KO) coculture system (Fig. 2k, l). However, other inflammatory factors, such as TNFα and IL-1β, were not significantly different between WT and Gal3 KO mice (Fig. 2k). When macrophages were taken from NC-fed mice, GSIS was not different between the Macs-WT and Macs-KO coculture systems (Supplementary Fig. 2k, l). However, when the macrophages used were from HFD-fed mice, GSIS in the MIN6-Macs-KO group was increased compared to that in the MIN6-Macs-WT group (Fig. 2l, m), indicating that Gal3 plays a critical role in acute macrophage-induced β-cell dysfunction.

## Gal3 knockout improves glucose hemostasis in HFD-fed and *db/db* mice

Since Gal3 significantly inhibited GSIS in vitro, we next assessed β-cell function and glucose homeostasis in WT, Gal3[+/−], and Gal3[−/−] mice. As expected, compared to levels in WT mice, the circulating Gal3 levels were approximately 50% lower and totally lost in Gal3[+/−] and Gal3[−/−] mice, respectively (Fig. 3a). The body weight and IPGTT results were comparable among the groups fed NC (Supplementary Fig. 3a, b). However, among HFD-fed mice, Gal3 deficiency led to more glucose tolerance combined with an increase in blood insulin secretion upon glucose stimulation, with no change in weight gain in mice fed a HFD (Fig. 3b–e and Supplementary Fig. 3c).

To further investigate the effect of Gal3 on β cell dysfunction, in vivo insulin secretion was induced with a standard hyperglycemia clamp. Blood glucose was clamped at a high level of glucose infusion and sustained at $14 \pm 0.5$ mM during the 120-min assay (Fig. 3f, Supplementary Fig. 3d, e). The infused glucose continued to stimulate insulin secretion in mice (Fig. 3g, Supplementary Fig. 3f, g). Notably, both the first and second phases of insulin release were significantly improved in Gal3[+/−] and Gal3[−/−] mice fed a HFD (Fig. 3g, h, Supplementary Fig. 3g, h), whereas these phases were comparable between Gal3-deficient and WT mice fed NC (Supplementary Fig. 3 f). Moreover, Gal3[+/−] mice fed a HFD and those fed NC were markedly more insulin sensitive than WT mice, which was manifested by an increased glucose infusion rate (GIR) (Fig. 3i and Supplementary Fig. 3i), with no change in Gal3[−/−] mice fed a HFD (Supplementary Fig. 3j). Furthermore, the islets from Gal3[+/−] mice fed a HFD exhibited a significant increase in GSIS (Fig. 3j). These results indicate that Gal3 knockdown ameliorates HFD-induced insulin secretion dysfunction.

Compared with HFD-fed mice, *db/db* C57BL/KS mice have more severe β-cell dysfunction and are a model for obesity-driven β-cell failure and T2DM[24]. The phenotype of diabetes in *db/db* mice appeared early and gradually became aggravated with increasing age[21]. To assess the role of Gal3 in *db/db* mice, we generated Gal3 KO-*db/db* (*db/db* GKO) mice (Fig. 3k). Plasma Gal3 levels significantly increased with the age of *db/db* mice (3.7-fold increase at 12 weeks old compared to that at 8 weeks old, Fig. 3l). As expected, both circulating and islet Gal3 were absent in *db/db* GKO mice (Fig. 3l, m). Importantly, in this model, we found that the phenotype of diabetes was markedly improved. For example, *db/db* GKO mice were much more insulin sensitive (Fig. 3n) and glucose tolerant (Fig. 3o and Supplementary Fig. 3k), with a lower body weight, which is probably due to the higher plasma glucagon levels and plasma glutathione (GSH) levels (Fig. 3p, Supplementary Fig. 3l, m). Moreover, *db/db* GKO mice featured improved first-phase insulin and C-peptide secretion starting at 9 weeks of age (Fig. 3q and Supplementary Fig. 3m) compared to *db/db* mice. Consistent with this response, the islets from *db/db* GKO mice exhibited an ~2-fold increase in GSIS (Fig. 3r), with comparable pancreas weights between the groups (Supplementary Fig. 3n). Taken together, these results indicate that Gal3 KO improves β-cell dysfunction and insulin resistance in *db/db* mice.

## Depletion of Gal3 derived from macrophages improves GSIS in HFD-fed mice

To further investigate the effect of Gal3 from macrophages, we depleted HFD-fed mice of macrophages by clodronate injection

 

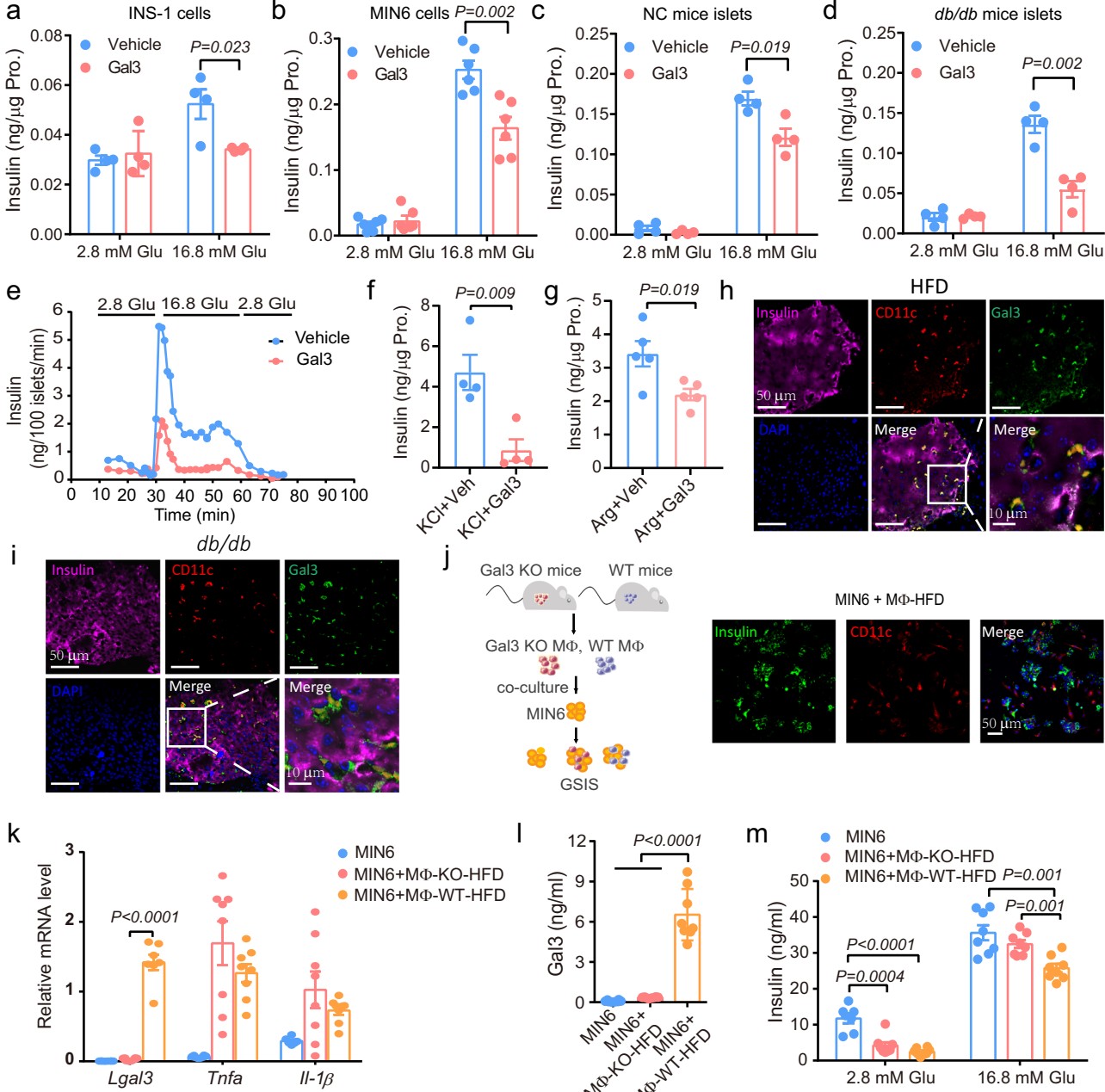

**Fig. 2 | Gal3 reduces islet and β-cell GSIS. a–d** Glucose-stimulated insulin secretion (GSIS) in INS-1 cells (**a**), MIN6 cells (**b**), NC mice primary islets (**c**), and *db/db* mice primary islets (**d**) with Gal3 treatment (MIN6 cells 80 ng/ml, INS-1 cells and islets 250 ng/ml) for 6 h. Gal3 existed all the time in the experimental process. ((**a, c, d**), *n* = 4 biologically independent cell samples; (**b**), *n* = 6 biologically independent cell samples.) **e** Islet perfusion experiment in islets from 12-week-old NC mice with Gal3 (250 ng/ml) treatment for 6 h. **f** KCl (30 mM) stimulated insulin secretion in mice islets with or without Gal3 (250 ng/ml, 1 h) treatment (*n* = 4 biologically independent cell samples). **g** Arginine (10 mM) stimulated insulin secretion in mouse islets with glucose (16.8 mM) and Gal3 (250 ng/ml, 1 h) treatment (*n* = 5 biologically independent cell samples). (Islets from 12-week-old NC mice (**c, e, f, g**) and from 17-week-old *db/db* mice (**d**)). **h–i** Confocal image of Gal3 colocalization with CD11c by immunostaining in insulin-positive cells in pancreatic sections from HFD-fed mice (**h**) or

from *db/db* mice (**i**). *n* = 6 islets in 3 pancreatic sections (HFD: 16 weeks from 8 weeks of age, *db/db* 12 weeks of age). **j** Sketch map of MIN6 cells and MΦ coculture system and immunofluorescence staining of MIN6 cells and MΦ in the coculture system. Insulin (Green) labelled MIN6 cells and CD11c (Red) labelled MΦ. The MΦ were peritoneal macrophages from WT or Gal3 KO mice on a HFD feeding 12 weeks. *n* = 3 independent experiments. **k** Proinflammatory genes expression in MIN6 and MΦ from HFD-fed mice coculture system (*n* = 6 for MIN6 group and *n* = 8 biologically independent cell samples for MIN6 + MΦ groups). **l** Gal3 levels in condition medium from coculture system with MΦ isolated from Gal3 KO or WT mice on a HFD. **m** GSIS in MIN6 and MΦ coculture system. *n* = 8 biologically independent cell samples (**l, m**). Data were analyzed by two-sided Student's *t*-test without adjustments for multiple comparisons. All data are presented as the mean ± SEM. Source data are provided as Source Data file.

(Fig. 4a). Immunofluorescence staining showed that clodronate treatment resulted in a dramatic deduction in macrophages and Gal3 in pancreatic islets (Supplementary Fig. 4a). There was also a marked decrease in Gal3 levels in the circulation, islets and supernatant of islets (Fig. 4b, Supplementary Fig. 4b, c), with a decreasing trend

decrease in body weight (Supplementary Fig. 4d), indicating that the majority of the increase in Gal3 in circulation and islets comes from macrophages. Moreover, macrophage depletion led to an increase in glucose tolerance and insulin secretion in vivo and in primary islets (Fig. 4c–f).

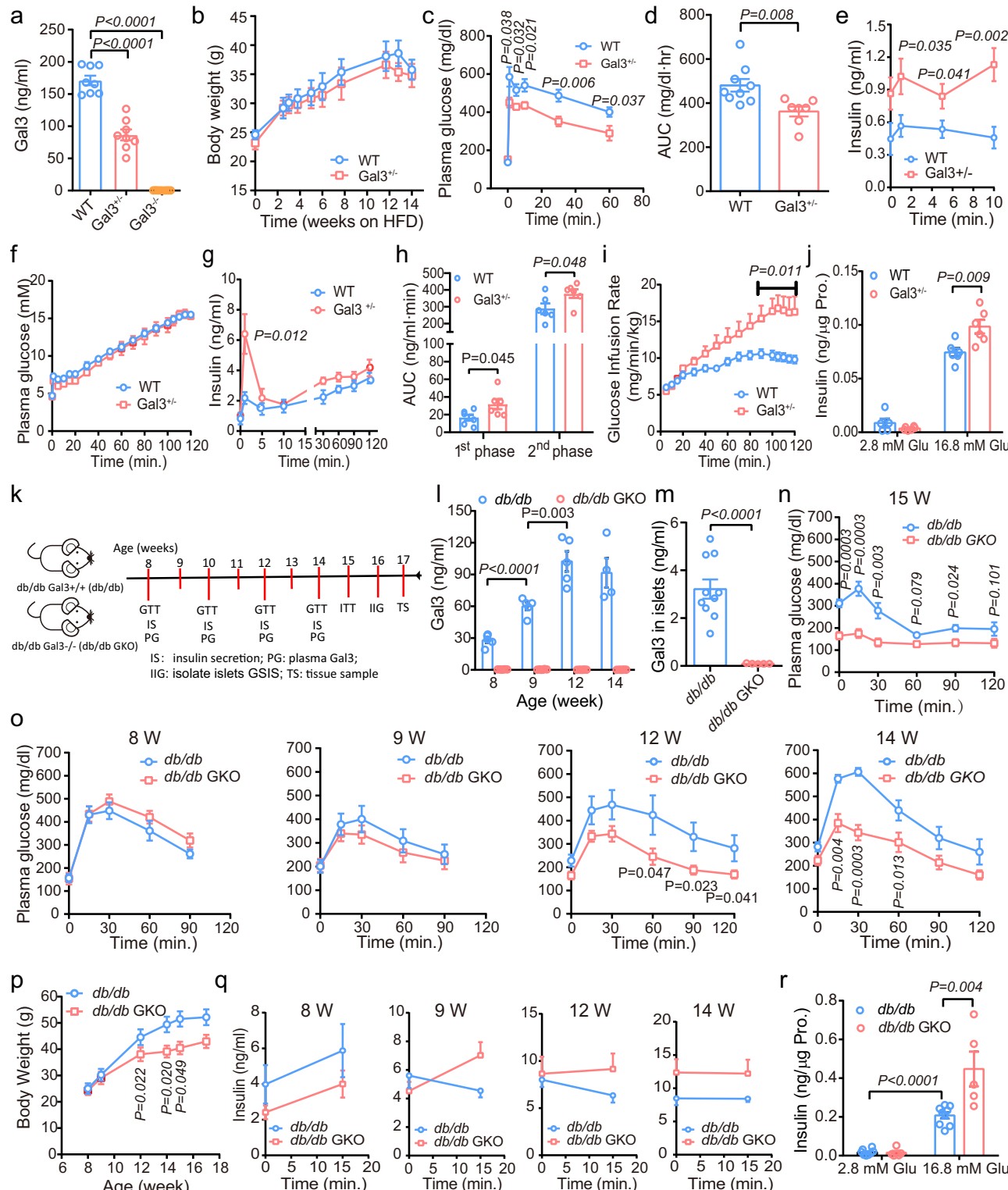

**Fig. 3 | Gal3 knockout improves diabetic phenotype in HFD-fed and *db/db* mice.** **a** Blood Gal3 levels in WT, Gal3+/− and Gal3−/− mice on HFD-fed (*n* = 8 mice). **b**–**e** Body weight (**b**, *n* = 12 WT mice, *n* = 7 Gal3+/− mice), intravenous glucose tolerance test (IVGTT) (**c**), Area under curve (AUC) during IVGTT (**d**) and first-phase insulin secretion (**e**) in WT and Gal3+/− mice on HFD-fed after 6 h of fasting. **c**–**e**, n = 9 WT mice, *n* = 7 Gal3+/− mice. **f**–**i** Plasma glucose (**f**), Insulin level (**g**), The AUC of the first phase of insulin secretion (from 0 to 10 min) and the second insulin secretion (from 10 to 120 min) (**h**), and GIR (**i**) in hyperglycemic clamp study in Gal3+/− mice on HFD-fed. *n* = 6 mice. **j** GSIS in primary islets from Gal3+/− mice on HFD-fed (*n* = 6 biologically independent samples). HFD: 8–12 weeks from 8 weeks of age (**c**–**j**).

**k** Experimental scheme of the *db/db* and *db/db* GKO mice. **l, m** Gal3 concentration in plasma (**l**, *n* = 5 *db/db* mice, *n* = 6 *db/db* GKO mice) and islets (**m**, *n* = 10 *db/db* mice, *n* = 5 *db/db* GKO mice). **n**–**r** ITT (**n**), IPGTT (**o**), Body weight (**p**), time course study of first-phase insulin secretion (**q**), and islets GSIS (**r**) in *db/db* and *db/db* GKO mice. *n* = 5 *db/db* mice, *n* = 7 *db/db* GKO mice (**n**–**q**); *n* = 8 (*db/db*), n = 5 (*db/db* GKO) biologically independent samples (**r**). The age of mice (**l**–**r**) was same with (**k**). Data were analyzed by two-sided Student's *t*-test without adjustments for multiple comparisons. All data are presented as the mean ± SEM. Source data are provided as Source Data file.

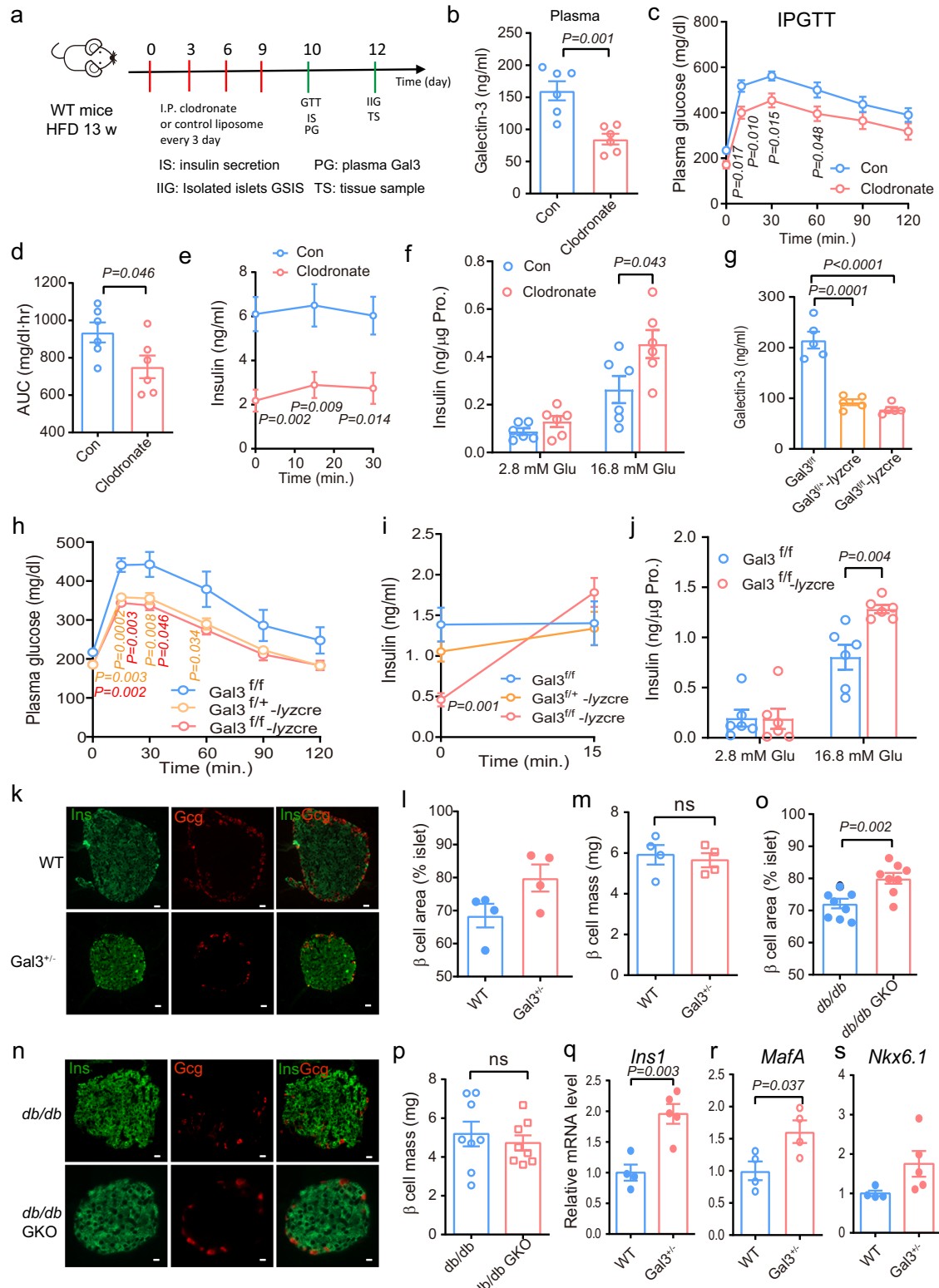

Furthermore, we generated macrophage Gal3 knockout mice with Cre-Loxp system (Supplementary Fig. 4e). As Fig. 4g showed, compared to levels in Gal3^f/f mice, the circulating pancreatic Gal3 levels were significantly lower in Gal3^f/f-*lyz* Cre mice on HFD (Supplementary Fig. 4g–f). Of note, macrophage-specific Gal3 depletion led to an improvement in glucose tolerance and insulin secretion during IPGTT in HFD-fed mice (Fig. 4h, i). In line with the in vivo data, in primary islets from Gal3^f/f-*lyz* Cre mice on HFD, glucose induced higher insulin secretion compared to Gal3^f/f group (Fig. 4j).

Together, these results indicate that Gal3 from macrophages plays an important role in β-cell dysfunction and glucose intolerance in HFD-fed mice.

**Histologic and gene expression studies of WT and Gal3 KO islets**
To determine whether Gal3 deletion affects the morphology of islets, we conducted immunofluorescence (IF) studies and found that islets from both groups were composed of a large insulin-positive β-cell core surrounded by punctate glucagon-positive α cells on the surface

**Fig. 4 | Depletion macrophages improve GSIS in HFD mice and histologic and gene expression studies of WT and Gal3 KO islets. a** Experimental scheme of HFD mice by clodronate injection. **b** Gal3 level in blood. **c–f** IPGTT (**c**), AUC of IPGTT (**d**), first-phase insulin secretion (**e**) and GSIS in primary islet from clodronate- or control-treated HFD mice (**f**). *n* = 6 mice (**b–e**). **g–j** Gal3 levels in the blood (**g**, *n* = 5 mice), IPGTT (**h**), first-phase insulin secretion (**i**), and GSIS in primary islets (**j**) in HFD-fed mice, HFD: 4 weeks from 8 weeks of age. *n* = 6 mice (Gal3^f/f^ and Gal3^f/+^ -*lyz*cre), *n* = 7 mice (Gal3^f/f^ -*lyz*cre) (**h**, **i**), *n* = 6 biologically independent samples (**f**, **j**). **k** Immunohistochemistry analysis of insulin (green) and glucagon (red) in pancreas of WT and Gal3^+/−^ mice on HFD-fed. Scale bar, 20 μm. **l** β-cell area in (**k**). **m** β-cell mass in WT and Gal3^+/−^ on HFD-fed mice. *n* = 4 islets in 4 pancreatic sections (HFD: 16 weeks from 8 weeks of age (**k–m**). **n** Immunohistochemistry analysis of insulin

(green) and glucagon (red) in pancreas of *db/db* and *db/db* GKO mice. Scale bar, 20 μm. **o** β-cell area of (**n**). **p** β-cell mass in *db/db* and *db/db* GKO mice. *n* = 8 islets in 5 pancreatic sections (16 weeks of age, **n–p**). **q–s** mRNA level of *Ins1* (**q**), *MafA* (**r**) and *Nkx6.1* (**s**) in islets from WT and Gal3^+/−^ mice on a HFD feeding (16 weeks feeding from 8 weeks of age). mRNA level of each gene was normalized to the *36B4* mRNA level in the same sample. *n* = 4 mice (WT in (**q–s**) and Gal3^+/−^ in **r**), *n* = 5 mice (Gal3^+/−^) (**q**, **s**). Mice information (**b–e**, **g–i**) was same with (**a**). Data were analyzed by two-sided Student's *t*-test without adjustments for multiple comparisons. All data are presented as the mean ± SEM. Source data are provided as Source Data file. *P*-values in red in (**h**) represent Gal3^f/f^-*lyz* Cre vs Gal3^f/f^ mice; *P*-values in orange in (**h**) represent Gal3^f/+^-*lyz* Cre vs Gal3^f/f^ mice.

(Fig. 4k, n). However, the islets were smaller in the Gal3^+/−^ mice fed a HFD than in the control mice (Fig. 4k), accompanied by a 16.6% increase in the β-cell area (% islet) and comparable β-cell mass, as measured by morphometric analyses of insulin-positive islet cells (Fig. 4l, m). Consistently, there was an 11% increase in the β-cell area (% islet) and comparable β-cell mass in *db/db* GKO mice (Fig. 4o, p). Additionally, HE staining showed that deficiency of Gal3 decreased vacuolar degeneration cells in islets (Supplementary Fig. 4h, i).

Since Gal3 affected the morphology of islets, we next measured the changes in pancreatic cell expression of genes related to β-cell function, dedifferentiation, and apoptosis. The results showed that genes involved in β-cell function, such as *Ins1*, *Mafa* and *Nkx6.1*, were increased in the islets and pancreas from HFD-fed Gal3^+/−^ mice and *db/db* GKO mice compared to controls (Fig. 4q–s, Supplementary Fig. 4j–l). However, Gal3 deficiency in mice or long-term Gal3 treatment (24 h) in MIN6 cells or human islets had no effect on genes involved in dedifferentiation (Supplementary Fig. 4m–o). Additionally, 48 h of treatment with Gal3 led to increased expression of the pro-apoptotic gene *Bax* and decreased expression of the anti-apoptotic gene *Bcl2* in MIN6 cells (Supplementary Fig. 4p, q).

Taken together, these results indicated that Gal3 deficiency in mice reduces apoptosis and normalizes islet morphology.

## Dysfunction of cytosolic calcium in Gal3-treated MIN6 cells and mouse islets

Given the magnitude of the effects of Gal3, we conducted detailed in vitro studies to elucidate the mechanisms by which Gal3 affects GSIS. Generally, GSIS is mediated by glucose uptake, glucose metabolism, calcium influx, etc. We examined glucose uptake and found that Gal3 had no effect on either glucose uptake or the expression of *Glut2*, the glucose transporter on β cells, in MIN6 cells or human islets (Supplementary Fig. 5a–c).

Gal3 binds to insulin receptor (IR) and blocks insulin signaling[7]. We next investigated whether IR mediates the effects of Gal3. MIN6 cells were pretreated with S961, an inhibitor of IR, followed by high-glucose challenge. Surprisingly, S961 did not decrease and actually increased GSIS in MIN6 cells (Supplementary Fig. 5d), with inhibition of AKT phosphorylation induced by insulin (Supplementary Fig. 5e). These results indicate that IR does not mediate the effect of Gal3 on GSIS in β cells.

We investigated cytosolic calcium signals in islets from mice with β-cell-specific expression of GCaMP6f, a protein encoding a Ca²⁺ indicator with green fluorescence. Gal3 dramatically inhibited the calcium transient triggered by 30 mM KCl (Supplementary Fig. 5f) or 10 mM arginine (Supplementary Fig. 5g). Next, we measured cytosolic calcium dynamic signals with the calcium dye Fluo-4 with or without Gal3 (80 ng/ml) treatment in MIN6 cells (Fig. 5a) and found that Gal3 treatment induced a different transient cytosolic calcium pattern, and the Gal3 treatment group featured increased numbers of calcium spikes in the falling phase compared with the control group (Fig. 5a, b). Similarly, in primary islets with GCaMP6f expression in β cells, Gal3 treatment greatly increased the number of calcium spikes and reduced

the peak value of calcium transients (Fig. 5c–e). Subsequently, resting cytosolic calcium was detected by Fura-2 in MIN6 cells. Neither the average resting level of Ca²⁺ (Fig. 5f) nor the distribution of cytosolic calcium levels (Fig. 5g) was different between the vehicle and Gal3 treatment groups grown in low-glucose (2.8 mM) KRHB medium. This evidence suggests that the Gal3-induced abnormal cytosolic calcium behaviors in pancreatic β cells might be associated with impaired calcium influx, which is mediated by L-type voltage-gated calcium channels (L-VGCCs) and strongly triggered by high glucose.

To further study the mechanism, we performed patch clamp experiments, which provide more direct evidence of calcium currents, in a single β cell in addition to time-lapse intracellular calcium imaging (Fig. 5h). First, we tested the stability of the system. As shown in Supplementary Fig. 4i, there were no obvious differences in the calcium current or calcium influx at 1, 3, 5, and 10 min by recording continuously with vehicle solution perfusion, indicating that the calcium channels in MIN6 cells were not rundown even after 3 cycles. With this system, we first recorded calcium currents in MIN6 cells in vehicle solution and then recorded the calcium currents after perfusion with 80 ng/ml Gal3 in vehicle solution for 5 min. Finally, the calcium currents were recorded again after vehicle solution perfusion for 5 min. We found that Gal3 could decrease the calcium current under different voltages (from −50 mV to 20 mV, +10 mV per step) in MIN6 cells (Fig. 5i, j), and the calcium current behaved similarly even after 5 min of washing, given that 5 min of washing may not remove the Gal3 that is already bound to proteins on the membrane (Fig. 4k).

Moreover, we found that nimodipine, an inhibitor of L-VGCCs, markedly and dose-dependently inhibited GSIS in MIN6 cells (Supplementary Fig. 5h). All these results demonstrated that the inhibitory effects of Gal3 on GSIS might occur through L-VGCCs.

## Gal3 binds to CACNG1 and affects insulin secretion

Next, we investigated the molecular mechanism by which Gal3 affects L-VGCCs. Considering the rapid inhibitory effect of Gal3 on the whole-cell Ca²⁺ current, we predicted that Gal3 affects the components of L-VGCCs in β cells through protein–protein interactions. Previous studies have shown that Gal3 binds to N-linked glycopeptides, such as N-linked (GlcNAc...) asparagine[25,26]. By analyzing the human protein–protein interactome established by Huttlin and his colleagues, we found that CACNG1, a key component of the γ subunits of L-VGCCs, interacted with Gal3[27–29]. Accordingly, we used co-IP experiments to confirm that GFP-tagged Gal3 precipitated with Flag/myc-tagged CACNG1 in both MIN6 and 293T cells (Fig. 6a, Supplementary Fig. 6a). Furthermore, there was an endogenous interaction in the MIN6 cell and macrophage coculture system and in islets from *db/db* and HFD-fed mice (Fig. 6b, c), but not in islets either from *db/m* or from NC-fed mice (Supplementary Fig. 6b), suggesting that the interaction is a pathological event instead of a physiological process. Colocalization occurred on the membrane and in the cytoplasm (Fig. 6d). The proximal ligation assay (PLA), an antibody-based method for detecting protein–protein interactions in situ (at distances <40 nm), suggested a direct interaction between Gal3 and CACNG1 (Fig. 6e). Finally, we

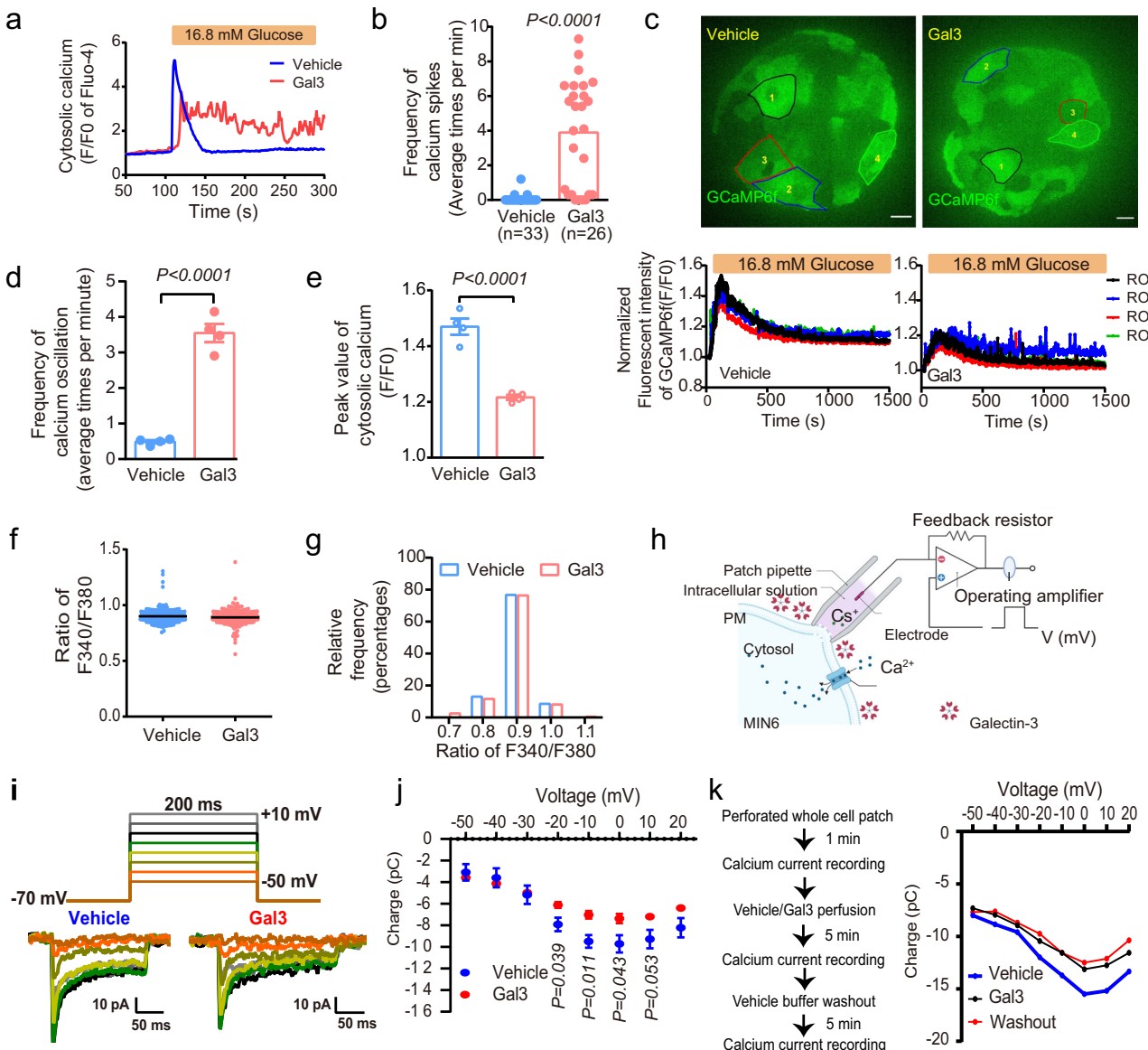

**Fig. 5 | Dysfunction of cytosolic calcium in Gal3-treated MIN6 cells and mouse islets. a** Typical traces of time-lapse calcium imaging via fluo-4, AM (1 µM) in MIN6 cells with high glucose (16.8 mM) stimulation and Gal3 (80 ng/ml, 6 h) treatment. $n = 33$ cells in vehicle group, and $n = 26$ cells in Gal3 group. **b** Frequency of cytosolic calcium spikes of (**a**). **c–e** Cytosolic calcium dynamics (**c**), frequency of cytosolic calcium spikes (**d**) and peak value of cytosolic calcium transient (**e**) stimulated by high glucose (16.8 mM) in different β cells of Ins1-GCaMP6f mouse islets with or without Gal3 (250 ng/ml, 6 h) treatment. Scale bar, 10 µm. $n = 4$ biologically independent samples (12-week-old NC-fed mice, **c–e**). **f, g** The ratios of fluorescence signals (**f**) and distribution analysis of concentrations of cytosolic free $Ca^{2+}$ ([$Ca^{2+}$]i)

(**g**) by fura-2, AM (2 µM) at 340 nm and 380 nm in MIN6 cells with basal glucose (2.8 mM) and Gal3 (80 ng/ml, 6 h) treatment. $n = 222$ cells in vehicle group, and $n = 232$ cells in Gal3 group. **h** Diagram of patch clamp experiment, created with BioRender.com. **i, j** $Ca^{2+}$ currents records in MIN6 cells without Gal3 and sequentially given Gal3 (80 ng/ml, 5 min) ($n = 4$ biologically independent cell samples). **k** $Ca^{2+}$ currents records in MIN6 cells without Gal3 and sequentially given Gal3 (80 ng/ml, 5 min) and then given vehicle buffer washout. Data were analyzed by two-sided Student's $t$-test without adjustments for multiple comparisons. All data are presented as the mean ± SEM. Source data are provided as Source Data file.

expressed and purified the extracellular domains of mouse CACNG1 (30-109 AA) with His tags and measured the equilibrium dissociation constant ($K_D$) of Gal3 with CACNG1 in a concentration gradient using a biolayer interferometry (BLI) system. The $K_D$ for the interaction was $1.8 ± 0.26E-07$ M (Fig. 6f), suggesting direct binding as well. In vitro Gal3-tracing experiment showed that in macrophages and MIN6 cells coculture system, GFP labeled Gal3 from macrophages colocalized with CACNG1 on MIN6 cells when the Gal3 KO macrophages were infected with adenovirus expressing Gal3-GFP (Ad-Gal3) (Fig. 6g). These results suggest that macrophage-released Gal3 could bind to CACNG1 on β cells.

It has been reported that CACNG1 possesses two N-glycosylation sites, N43 and N80[30,31]. A single-amino-acid mutation (N43A or N80A) of CACNG1 slightly decreased the interaction of Gal3 with CACNG1, while the N43A/N80A double mutation almost completely blocked the interaction (Fig. 6h). The CACNG family contains eight subunits. Real-time PCR revealed that the top three expressed genes in islets were *Cacng1*, *Cacng6*, and *Cacng8*, and these genes were almost comparable in islets between *db/db* mice and controls (Supplementary Fig. 6c). Only CACNG1 but not CACNG6 or CACNG8 interacted with Gal3 (Supplementary Fig. 6d). The co-IP results showed a weak interaction of Gal3 with other subunits, including CACNG2, CACNG3, CACNG4,

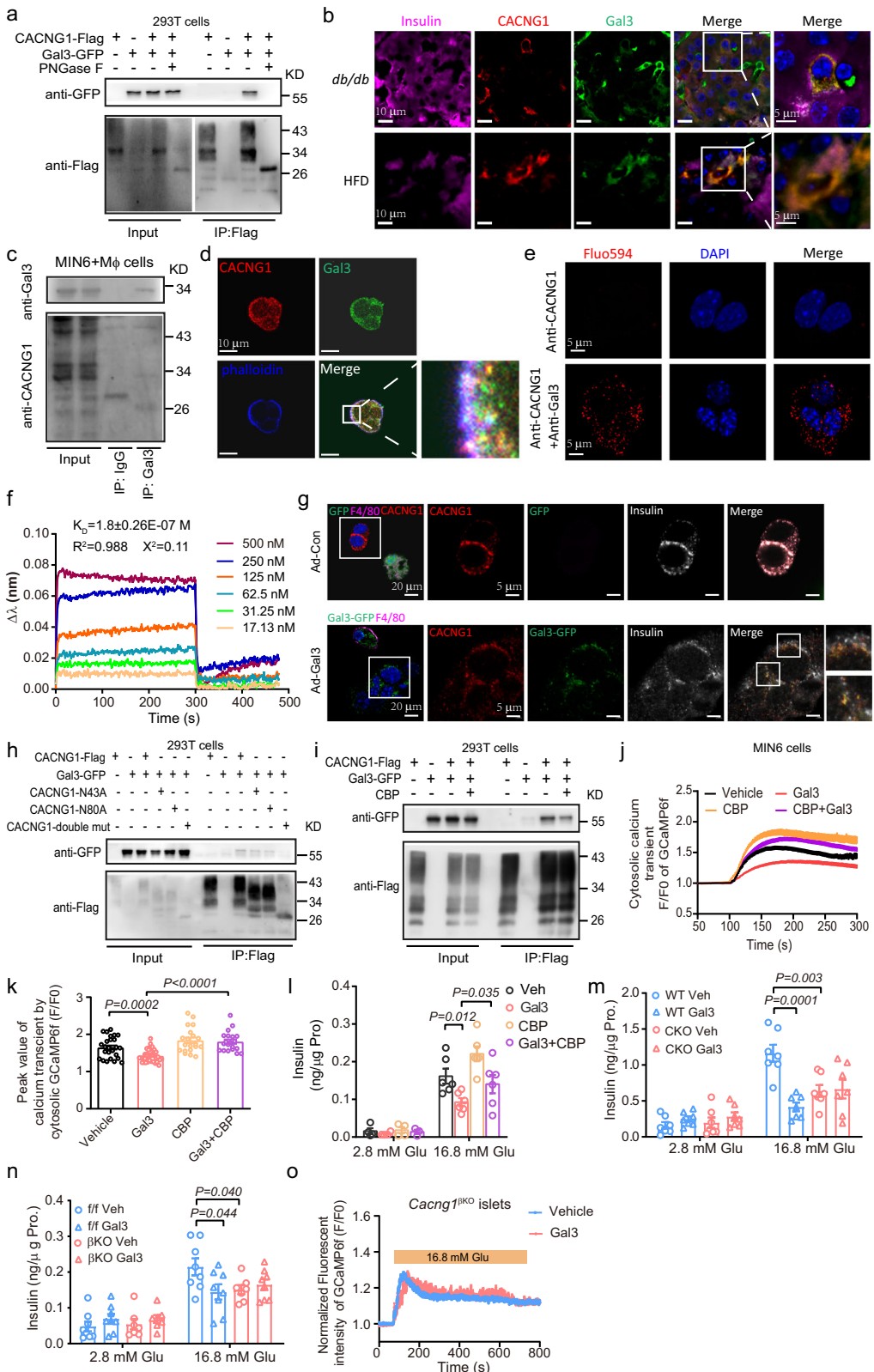

CACNG5, and CACNG7 (Supplementary Fig. 6e–g). Since there was less expression of *Cacng2-Cacng7*, we speculated that *Cacng1* mediates the effect of Gal3. To test this hypothesis, we employed an extracellular blocking peptide of CACNG1 (CBP), which blocks CACNG1 amino acids 30 to 44, and found that CBP significantly decreased the interaction of Gal3 and CACNG1 (Fig. 6i and Supplementary Fig. 6h). Most importantly, CBP reversed the effect of Gal3 on the cytosolic calcium

transient trace and peak values in MIN6 cells and GSIS levels in mouse islets (Fig. 6j–l).

As shown in Fig. 6g, h, CACNG1-Flag produced a smear band. Treatment with PNGase F, which hydrolyzes N-linked oligosaccharides in glycoproteins, gave a clear band and abolished the interaction (Fig. 6a). These data suggested that Gal3 binds to glycosylated CACNG1.

**Fig. 6 | Gal3 binds to CACNG1 and affects insulin secretion. a** Co-immunoprecipitation of Gal3-GFP and Flag-tagged CACNG1 (*n* = 3 independent experiments). **b** Colocalization of Gal3 and CACNG1 in islets from *db/db* (17 weeks of age) and HFD (16 weeks feeding from 8 weeks of age) mice, *n* = 3 islets in 3 pancreatic sections. **c** Co-immunoprecipitation of Gal3 and CACNG1 in MIN6 and macrophage coculture system. **d** Gal3 and CACNG1 were colocalization on the membrane in MIN6 cells. **e** Direct interaction of Gal3 and CACNG1 in the PLA. Scale bar, 5 μm. **f** The Gal3 and CACNG1 binding affinity assay was performed using a BLI system. **g** Gal3-tracing approach. Scale bar, 20 μm and 5 μm. **h** Co-immunoprecipitation of Gal3-GFP and Flag-tagged WT CACNG1, single amino acid mutant CACNG1 (N43A), CACNG1 (N80A), and double amino acid mutant CACNG1 (N43A/N80A). **i** Co-immunoprecipitation of Gal3-GFP and Flag-tagged CACNG1.

*n* = 1 (**c**), *n* = 3 (**d**, **e**, **g**–**i**) independent experiments. **j** Effects of Gal3 (80 ng/ml, 30 min) and CBP (30 min) on high glucose (16.8 mM) stimulated calcium signals. **k** Peak value of cytosolic calcium transients in (**j**). *n* = 20 cells (**j**–**k**). **l** GSIS in islets with Gal3 (250 ng/ml, 1 h) and CBP (1 h) treatment. **m** GSIS in islets from WT and CKO mice with or without Gal3 (250 ng/ml, 1 h) treatment. **n** GSIS in Cacng1^f/f and *Cacng1* Ins1Cre-GCaMP6f (*Cacng1*^βKO) mice islets with or without Gal3 (250 ng/ml, 1 h) treatment. *n* = 6 (**l**), *n* = 7 (**m**), *n* = 8 (**n**) biologically independent samples). Cytosolic calcium dynamics in β cells from Cacng1^βKO mice islets with or without Gal3 (250 ng/ml, 1 h) (*n* = 25 cells). Islets are from 10-week-old NC mice (**l**–**o**). Data were analyzed by two-sided Student's *t*-test without adjustments for multiple comparisons. All data are presented as the mean ± SEM. Source data are provided as Source Data file. CBP, CACNG1 blocking peptide (1 μg/ml).

## CACNG1 KO improves β cell function and glucose intolerance

To confirm the role of CACNG1 in the process by which Gal3 inhibits GSIS, we generated global *Cacng1* knockout (CKO) mice by using the CRISPR/Cas9 system (Supplementary Fig. 6i). qPCR analysis revealed a 96% reduction in *Cacng1* expression in the islets of CKO mice compared to the expression levels in WT mice (Supplementary Fig. 6j). And the CACNG1 protein levels were also reduced in the islets of CKO mice (Supplementary Fig. 6k–l). In islets from CKO mice, GSIS was decreased to the same extent as in the Gal3 treatment, and Gal3 could not further lower GSIS in islets from CKO mice (Fig. 6m), with no effect on intracellular insulin levels (Supplementary Fig. 6m). Moreover, we also generated β-cell-specific knockout *Cacng1* and β-cell-specific knock-in genetically encoded calcium indicator GCaMP6f mice (*Cacng*1 flox/flox, Ins1 cre, GCaMP6f flox/flox, *Cacng1*^βKO) (Supplementary Fig. 6n-o). Similar to the global *Cacng1* KO mouse islets, GSIS was reduced in islets from *Cacng1*^βKO mice compared to *Cacng1*^f/f mice, and in *Cacng1*^βKO mouse islets, there was no further decrease in GSIS after Gal3 treatment (Fig. 6n). Moreover, cytosolic calcium transients stimulated by high glucose were attenuated in *Cacng1*^βKO islets, and Gal3 did not exhibit a dramatic inhibitory effect role on the peak value of calcium transients from Cacng1^βKO islets (Fig. 6o). These mice were also characterized by glucose intolerance (Supplementary Fig. 6p). These results suggested that CACNG1 was critical for the effect of Gal3 on GSIS.

In summary, we demonstrated that Gal3 induces abnormal cytosolic calcium responses via high-glucose stimulation and subsequent modulation of GSIS and that CACNG1 plays an important role in the effect of Gal3.

## Inhibition of Gal3 reverses β-cell dysfunction

The above research clearly shows that Gal3 can directly inhibit β cell GSIS in vitro and that genetic Gal3 loss of function improves GSIS and normalizes islet morphology in both HFD-fed and *db/db* mice. To further assess the potential translational implications, we applied a pharmacologic loss-of-function approach. We treated MIN6 cells with different Gal3 inhibitors, including GB1107[32], TD139[33] and compound 47 (Cpd47)[34], and found that all of these inhibitors improved GSIS (Fig. 7a, b). Consistent with this finding, administration of a Gal3-neutralizing antibody also improved GSIS in MIN6 cells (Fig. 7c). Furthermore, GB1107 significantly reversed the effect of Gal3 on the cytosolic calcium transient trace and peak values in MIN6 cells (Fig. 7d, e). To confirm these findings in vivo, *db/db* mice were treated with GB1107 (10 mg/kg) daily for approximately 4 weeks (Fig. 7f). As shown in Fig. 7g–k, GB1107 improved glucose tolerance and insulin resistance, with enhanced insulin secretion but no change in body weight. Additionally, the isolated primary islets from the treatment group showed improved GSIS (Fig. 7l). Similar results were also obtained in *db/db* mice injected with Gal3 antibody (200 μg/kg). As shown in Fig. 7m–p and Supplementary Fig. 7a, b, Gal3-antibody improved glucose tolerance and insulin secretion with reduction in body weight and food intake in *db/db* mice. Taken together, these data

in Gal3 inhibitors and Gal3 antibody support the idea that the blockade of Gal3 could be promising treatment for β-cell dysfunction and diabetes.

It is known that there is compensatory enlargement of the islets of *db/db* mice. Histologic analysis showed that GB1107 treatment made islets smaller, with no change in the pancreas mass compared to that of controls (Supplementary Fig. 7c–e). Consistently, the expression of *Ins1* increased with GB1107 treatment, whereas *Mafa* or *Gcg* expression was not different between the two groups (Supplementary Fig. 7f). Taken together, our results show that pharmacological inhibition of Gal3 improves islet morphology.

## Gal3 impairs GSIS in human islets

Gal3 had a strong effect on GSIS in mice, and we sought to confirm this effect in human islets. In agreement with the findings in obese and diabetic mice (Fig. 1g, h), *LGALS3* mRNA levels were increased approximately 4-fold in islets from diabetic patients compared with healthy subjects (Fig. 7q). Furthermore, Gal3 (from 80 to 250 ng/ml) decreased GSIS in human islets from either healthy persons or patients with diabetes (Fig. 7r, s). Treatment with the Gal3 inhibitor Cpd47 reversed the inhibition of GSIS by Gal3 in human islets (Fig. 7t). Together, these results indicate that Gal3 impairs β-cell function in humans.

## Discussion

Here, we report that Gal3 levels are elevated in the islets of mice and subjects with obesity and diabetes. Gal3 causes in vitro and systemic β-cell dysfunction. Here, Gal3 acutely inhibited GSIS. We also showed that Gal3, but not other inflammatory factors, released by macrophages acutely causes β-cell dysfunction. These effects occur through Gal3 combining with CACNG1, a subunit of L-VGCCs (Supplementary Fig. 7g). Thus, these data strongly argue for a role of Gal3 in β-cell dysfunction and the development of diabetes.

Previous studies have evaluated the effects of Gal3 in lung fibrosis, cardiovascular diseases, nephropathy, obesity, insulin resistance, and hepatic steatosis[35–38]. Additionally, some reports have shown an effect of Gal3 on pancreatic β-cell apoptosis in diabetic mice[19,39,40]. Consistent with this information, Gal3 treatment (48 h) increased the expression of genes encoding pro-apoptosis factors and decreased the expression of genes encoding anti-apoptosis factors in MIN6 cells (Supplementary Fig. 4p, q). These results reflect the long-term effect of Gal3 on apoptosis. Here, we discovered direct and transient effects of Gal3 on GSIS.

Gal3 is mainly produced and secreted by monocytes/macrophages[17], and our study showed that the Gal3 from macrophages plays an important role in β cell dysfunction since deletion of Gal3 in macrophages either through clodronate treatment or macrophage-specific KO greatly improved β-cell function in vivo. Gal3 is also expressed in other cell types in the cytoplasm, nucleus or on the cell surface[39,41]. Regarding the role of Gal3 in β-cell dysfunction, there are some conflicting data. For example, in pancreatic islets exposed to IL-1β, Gal3 is the most upregulated protein, and its overexpression

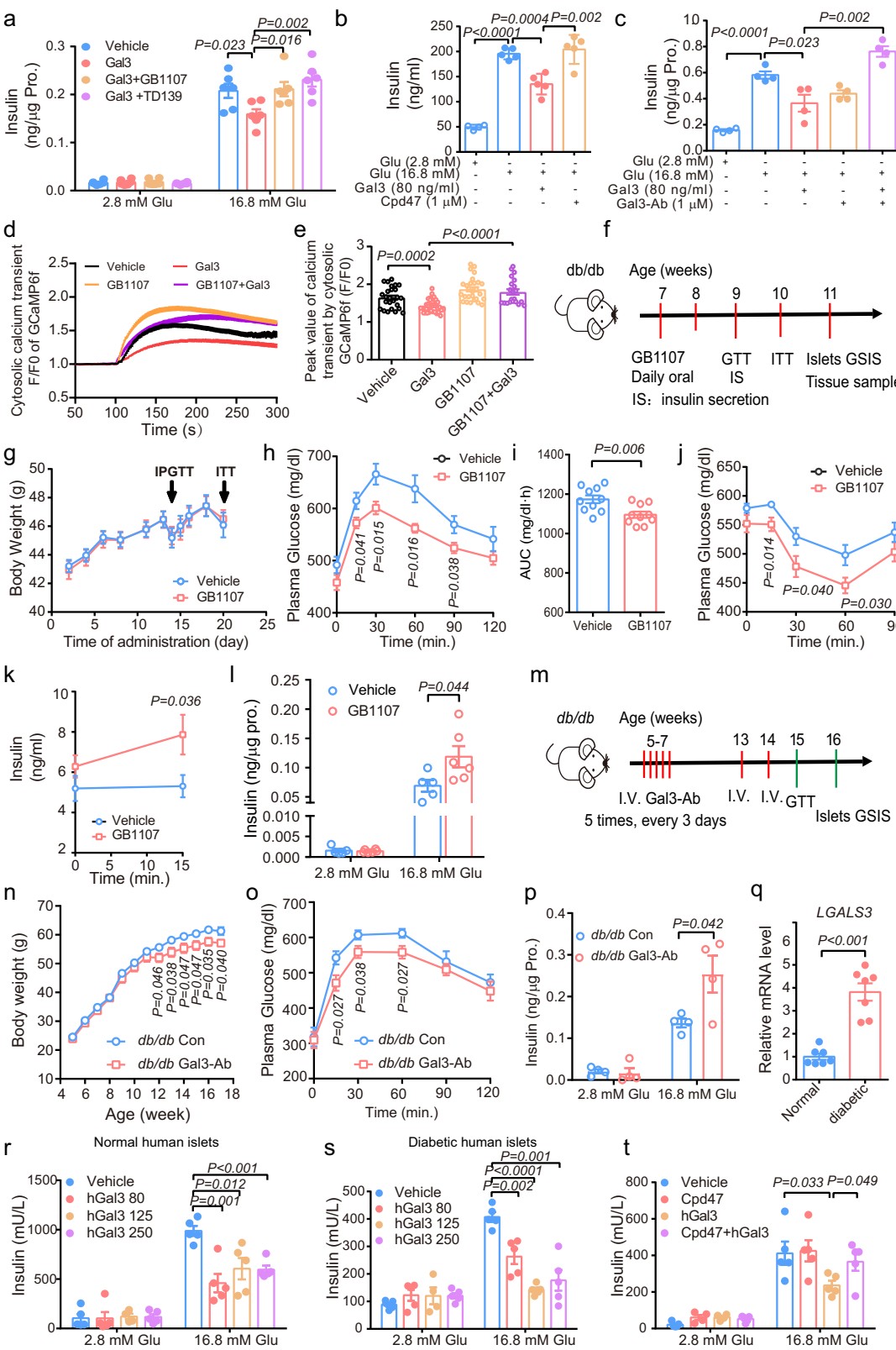

protects β cells from the cytotoxic effect of IL-1β[39], transgenic over-expression of Gal3 in pancreatic β cells was found to protect β cells from apoptosis in T1DM mice[40]. Of note, in these two reports, the increased Gal3 was overexpressed in β cells, namely, intracellular Gal3. In our studies, extracellular Gal3 impaired β-cell function. Intracellular and extracellular Gal3 may provide contradictory results, as shown in T cells[42,43]. Pejnovic et al.[44] showed that Gal3 ablation was associated

with increased inflammation in pancreatic islets from mice fed a HFD. However, in their studies, the body weight of WT mice was less than 30 g after 18 weeks of 60% HFD feeding, and their Gal3 KO mice gained approximately 5 g of body weight, which is rather inexplicable since long-term 60% HFD feeding significantly increased the weight of male WT mice in most reports. The inflammation of pancreatic islets is related to the body weight of HFD-fed mice to some extent. Therefore,

**Fig. 7 | Inhibition of Gal3 improves GSIS and glucose homeostasis in *db/db* mice. a–c** Effect of Gal3 (80 ng/ml, 1 h) and Gal3 inhibitors GB1107 (1 µM, 1 h), TD139 (1 µM, 1 h) (**a**) and Cpd47 (1 µM, 1 h) (**b**) and Gal3 neutralizing antibody (Ab) (1 µM, 1 h) (**c**) on GSIS in MIN6 cells. *n* = 6 (**a**), *n* = 5 (**b**), *n* = 4 biologically independent cell samples). **d** Effects of Gal3 (80 ng/ml, 30 min) and GB1107 (1 µM, 30 min) on high glucose (16.8 mM) stimulated calcium signals in MIN6 cells. **e**, Peak value of cytosolic calcium transient in MIN6 with or without Gal3 (80 ng/ml, 30 min) and GB1107 (1 µM, 30 min). *n* = 20 cells (**d**, **e**). **f–l** Experimental scheme (**f**), body weight (**g**), IPGTT (**h**), AUC of GTTs (**i**), ITT (**j**), in vivo GSIS (**k**), and GSIS of isolated primary islets (**l**) in *db/db* mice treated with GB1107 (10 mg/kg). *n* = 10 mice (**f–k**), *n* = 5 Vehicle and *n* = 6 GB1107 biologically independent samples (**l**). **m** Experimental scheme of *db/db* mice by Gal3 antibody injection. At the age of 5 weeks, *db/db* mice were intravenously injected with Gal3 antibodies every 3 days

for a total of 5 consecutive injections. Additional injections were administered at the 13th and 14th weeks. **n–q** Body weight (**n**, *n* = 10 mice), IPGTT (15 weeks of age) (**o**, *n* = 9 mice), and GSIS in primary islets (16 weeks of age) (**p**, *n* = 4 biologically independent samples) from *db/db* mice. **q** *LGALS3* gene expression in primary islets from healthy persons and diabetic patients (*n* = 7 biologically independent samples). **r**, **s** GSIS in human primary islets from healthy persons (**r**) and diabetic patients (**s**) with hGal3 (250 ng/ml, 6 h) treatment. **t** GSIS in islets from healthy persons after hGal3 (250 ng/ml, 6 h) and Gal3 inhibitor Cpd47 (1 µM, 6 h) treatment. *n* = 5 (**r–t**) biologically independent samples. Data were analyzed by two-sided Student's *t*-test without adjustments for multiple comparisons. All data are presented as the mean ± SEM. Source data are provided as Source Data file. Mice information (**g–l**) was same with (**f**).

in the report of Pejnovic et al.[44], inflammation of pancreatic islets was increased in Gal3 KO mice fed a HFD.

Macrophages in pancreatic tissue are divided into two types, namely, intra-islet macrophages and peripheral (extra-islet) macrophages, and obesity or diabetes increases the numbers of intra-islet macrophages[9]. Consistently, our studies showed that the number of inflamed macrophages (CD11c-positive) in islets was dramatically increased in both HFD-fed and *db/db* mice compared to controls (Fig. 1c, d). Interestingly, the islets from *db/db* mice were characterized by more macrophage accumulation than those from HFD-fed mice (Fig. 1c, d). This may explain why the pancreases of *db/db* mice showed more adverse characteristics than those of HFD-fed mice.

It has been reported that the proinflammatory factors TNFα and IL-1β are increased and are involved in β-cell dysfunction in obesity and diabetes[14,22,45,46]. In previous studies, long-term (16, 24, and even 48 h) treatment with TNFα or IL-1β was used, while in our research, Gal3 impaired calcium currents within 5 min (Fig. 5i, j) and decreased GSIS within 1 h (Supplementary Fig. 2d). Therefore, in contrast to TNFα or IL-1β, the effect of Gal3 is acute. In addition, in the macrophage and MIN6 cell coculture experiment, the KO macrophage system did not cause MIN6 cell dysfunction, even though there was almost equal expression of TNFα and IL-1β between the WT and KO groups. Thus, our data suggest that Gal3 has an acute effect and plays a critical role in macrophage-induced β-cell dysfunction.

Previous studies have reported that cell–cell contact is required for macrophages to impact β-cell function in unstressed conditions, rather than some soluble factors released from macrophages[9]. This idea is in contrast with our finding that Gal3 impairs β-cell dysfunction. This discrepancy may be due to the experimental design. In Ying's study, after 24 h of incubation with macrophages or incubation in conditioned media, the authors washed MIN6 cells twice and incubated them overnight with 2.8 mM glucose DMEM, followed by the GSIS experiments. In other words, CM or cytokines secreted from macrophages were removed from the system during GSIS. Our study shows that Gal3 needs to be constantly present during GSIS to have an effect (Supplementary Fig. 2e). Accordingly, in our coculture experiment, MIN6 cells and macrophages were grown together, and we found an inhibitory effect of Gal3 (Fig. 2m).

In addition to the acute effect of Gal3 on insulin secretion, Gal3 may chronically affect β cell function. In HFD-fed and *db/db* mice, KO of Gal3 was accompanied by a significant increase in the β-cell area and the expression of genes related to β-cell function compared to those seen in controls (Fig. 4k–s). This result is consistent with the previous finding that Gal3 induces β-cell apoptosis and that targeted disruption of the Gal3 gene results in decreased susceptibility to streptozotocin-induced diabetes in mice[47]. The chronic effect of Gal3 may also be associated with calcium transients since calcium plays an important role in apoptosis[48–50].

In addition to genetic KO of Gal3, pharmacologic inhibition of Gal3 with inhibitors or antibody also induced remarkable therapeutic effects in diabetes. Blockade of Gal3 with inhibitors or antibody

reversed Gal3-mediated inhibition of GSIS in MIN6 cells (Fig. 7a, b). GB1107 treatment improved insulin secretion, glucose tolerance, and insulin sensitivity in *db/db* mice (Fig. 7h–l). At present, there are several Gal3 inhibitors in clinical trials for the treatment of nonalcoholic steatohepatitis (NASH), liver cirrhosis, portal hypertension or tumors[36,51,52], indicating that Gal3 is a druggable target.

In summary, we demonstrate that Gal3 is an important factor that impairs β-cell function in obesity and diabetes. We used both genetic (Gal3 KO mice, macrophage Gal3 KO, and Gal3 KO *db/db* mice) and pharmacologic (Gal3 inhibitors and antibody) approaches to investigate the phenotype, and all experiments provided consistent results. Overall, our findings reveal an effect of Gal3 on β cells: the transient action of Gal3 can directly inhibit GSIS in β cells; to accomplish this, Gal3 interacts with the regulatory subunit of L-VGCCs, impairs calcium currents and decreases $Ca^{2+}$ concentrations in β cells. These findings indicate that inhibition of Gal3 could be a therapeutic strategy for treating β-cell dysfunction and T2DM.

## Methods
### Study approval
All experiments using animals were performed following protocols approved by the Animal Experimentation Ethics Committee of the Chinese Academy of Medical Sciences, and all procedures were conducted following the guidelines of the Institutional Animal Care and Use Committees of the Chinese Academy of Medical Sciences. All animal procedures were consistent with the Animal Research: Reporting of In Vivo Experiments (ARRIVE) guidelines. All the methods and protocols for isolating and using human islets approved by the Medical Ethics Committee of Tianjin First Center Hospital[53] (No.2016N086KY) and written informed consent was obtained from every individuals. The donor-related information is available in Supplementary Table 6.

### Animals
Animals were housed in an animal facility on a 12 h/12 h light/dark cycle with a temperature at 21–23 °C and relative humidity at 50–60% with free access to food and water (chow diet, Beijing HFK Bioscience, 1035). Gal3$^{-/-}$ mice were kindly provided by Dr. Jerrold M. Olefsky from UC San Diego. *db/+* mice were provided by Dr. Huang from Wenzhou Medical University. Gal3$^{+/-}$ mice were acquired by breeding Gal3$^{-/-}$ mice with WT C57BL/6 J mice. WT, Gal3$^{+/-}$ and Gal3$^{-/-}$ littermates were produced by crossing Gal3$^{+/-}$ mice together. Gal3$^{+/-}$ *db/+* mice were acquired by breeding Gal3$^{-/-}$ mice with *db/+* mice. Gal3$^{-/-}$ *db/db* and *db/db* mice were produced by crossing Gal3$^{+/-}$ *db/+* mice together. Ins1 cre-GCaMP6f mice were provided by Dr. Liangyi Chen's lab at Peking University. Ins1 cre-GCaMP6f mice were generated by crossbreeding Ins1(Cre) mice[54] (Jackson Laboratories, strain #026801) and GCaMP6f fl/fl lines (Jackson Laboratories, strain # 029626) to obtain β-cell specific GCaMP6f, a protein encoding a $Ca^{2+}$ indicator with green fluorescence. Ten- to 14-week-old male Ins1-GCaMP6f mice were used for the islet β-cell calcium imaging

experiments. *Cacng1* global KO mice (background C57BL/6JGpt, strain #T029821) and *Cacng1*^fl/fl mice (background C57BL/6JGpt, strain #T020460) were purchased from Gempharmatech Co., Ltd. β-Cell-specific GCaMP6f-*Cacng1*^βKO mice were acquired by breeding *Cacng1*^fl/fl-GCaMP6f^fl/fl mice with *Cacng1*^fl/fl Ins1-cre^+/− GCaMP6f^fl/fl mice. *C57BL/6JGpt-Lyz2em1Cin(iCre)/Gpt* mice were purchased from Gempharmatech Co., Ltd (background C57BL/6JGpt, strain #T003822). ZK-003CKO (Gal3^f/f) mice were obtained from BIOCYTOGEN. Macrophage specific Gal3 knockout mice (Gal3^f/+-*lyz* Cre and Gal3^f/f-*lyz* Cre) were acquired by breeding Gal3^f/f mice and *Lyz2*-icre mice.

Eight-week-old male C57BL6/J (WT) mice and Gal3^−/− and Gal3^+/− mice were fed a high-fat diet (HFD, 60% of kcal from fat; D12492, Research Diets) for 8–16 weeks (Macrophage specific Gal3 knockout mice fed HFD for 4 weeks). Plasma Gal3 levels were measured by an ELISA from Aviscera Bioscience. Plasma insulin levels were measured with ELISA kits from ALPCO.

Male mice were used in this study. All *db/db* mice were BKS background.

Mice were euthanized with carbon dioxide at the designated time points during the study as per protocol approved by the Animal Experimentation Ethics Committee of the Chinese Academy of Medical Sciences.

## Cell culture

The mouse pancreatic β-cell line MIN6 was kindly provided by Prof. Xu from the Institute of Biophysics, Chinese Academy of Sciences. The rat pancreatic β-cell line INS-1was obtained from ACCEGEN (ABC-TC232S). Human embryonic kidney 293T (HEK293T) cells were purchased from the cell culture center of Peking Union Medical College. HepG2 cells were purchased from ATCC. FreeStyle 293F cells were provided by Prof. Wu from the Institute of Materia Medica, Chinese Academy of Medical Sciences and Peking Union Medical College. MIN6 and INS1 were authenticated by the cell morphology and function. HEK293T, HepG2 and FreeStyle 293F were authenticated by STR profiling. MIN6 cells were cultured in DMEM (Gibco, USA) with 15% v/v FBS (Gibco, USA), INS-1 cells were cultured in RPMI 1640 (Gibco, USA), with 10% FBS, 10 mM HEPES, and 1 mM sodium pyruvate. Both MIN6 and INS-1 cells were cultured with 100 U/ml penicillin/streptomycin (Gibco, USA), and 50 μM β-mercaptoethanol (Sigma–Aldrich, USA). HEK293T and HepG2 cells were cultured in DMEM medium (Gibco, USA) supplemented with 10% FBS. FreeStyle 293F cells were cultured in FreeStyle 293 Expression Medium (Thermo Fisher Scientific, 12338018), with 1% FBS and shaking. The cells were incubated at 37 °C in a suitable atmosphere containing 95% $O_2$ and 5% $CO_2$.

## Primary islet isolation and culture

Human islets were provided by Tianjin First Central Hospital. Islet isolation was performed as previously described[55]. All the experimental protocols including any relevant details were approved by the Medical Ethics Committee of Tianjin First Center Hospital. In general, the pancrease were distended with Serva NB1 enzymes (Serva, Germany) after trimming, then the tissue was digested using the modified Ricordi semi-automatic method for 8–10 min and washed and collected for further purification. After incubation in the University of Wisconsin (UW) solution for 30 min, the digested tissue was purified using University of Illinois at Chicago UW/Biocoll method (UIC-UB) gradient in a Cobe 2991 cell separator (Cobe 2991, Cobe, USA). Finally, the high purity islets (more than 70%) were collected and cultured in CMRL culture media (Mediatech, USA) containing IGF (Cell Sciences, USA) and 20% human albumin (CSL Behring GmbH, Germany) at 37 °C in 5% $CO_2$ incubator. To isolate primary mouse islets, the bile duct near the ampulla of Vater was ligated, and the common bile duct was cannulated and injected with 3 ml of D-Hank's buffer containing collagenase V (0.5 mg/ml; Sigma, USA). The pancreas was dissected from the surrounding tissues, removed, and incubated in a stationary bath

for 10 min at 37 °C. The tissues were vibrated and filtered through a 600 μm cell filter. The samples were centrifuged at 4 °C and washed with Hank's buffer. Primary islets were handpicked under a stereoscopic microscope, and then were incubated in complete RPMI 1640 medium at 37 °C in a humidified 5% $CO_2$ atmosphere.

## Glucose-stimulated insulin secretion (GSIS) assays

Static GSIS experiments were performed as described previously[56]. Briefly, the cells or islets were washed with 2.8 mM glucose Krebs Ringer Bicarbonate buffer (KRB buffer; 2.5 mM CaCl$_2$·2H$_2$O, 1.2 mM MgSO$_4$·7H$_2$O, 1.2 mM KH$_2$PO$_4$, 4.8 mM KCl, 119 mM NaCl, and 25 mM NaHCO$_3$, supplemented with 10 mM HEPES and 0.1% BSA (Invitrogen, USA)), and after fasting for 60 min with 2.8 mM glucose KRB buffer, the cells were incubated for 60 min in fresh 2.8 mM or 16.8 mM glucose KRB buffer. Insulin concentrations in the supernatant and the cellular insulin content were determined using ultrasensitive mouse ELISA kits (Alpco, USA). For INS-1, insulin levels were determined using ultrasensitive rat ELISA kits (Alpco, USA). The information of the ELISA kits are listed in Supplementary Table 1.

## Gal3 treatment of GSIS

For the 6 h treatment, the cells/islets were first treated with Gal3 (40, 80, 125 or 250 ng/ml) in culture medium for 4 h, washed twice with 2.8 mM glucose KRB buffer, and then fasted in 2.8 mM glucose KRB buffer for 1 h with Gal3 (40, 80, 125 or 250 ng/ml), followed by incubation in fresh 2.8 mM or 16.8 mM glucose KRB buffer for 1 h with Gal3 (40, 80, 125 or 250 ng/ml). For 1 h treatment, Gal3 (80 ng/ml) was only present in the last 1 h of GSIS. Insulin, glucagon and somatostatin concentrations in the supernatant were determined by ELISA kits.

## KCl-stimulated insulin secretion assays

Mouse islets were washed with fresh 1.0 mM glucose KRB buffer. After fasting for 60 min with 1.0 mM glucose KRB buffer, cells were incubated for 60 min in fresh 1 mM glucose KRB buffer with KCl (30 mM) and with or without Gal3 (250 ng/ml). Insulin concentrations in the supernatant were determined.

## Arginine-stimulated insulin secretion assays

Mouse islets were washed with fresh 1.0 mM glucose KRB buffer. After fasting for 60 min with 1.0 mM glucose KRB buffer, cells were incubated for 60 min in fresh 16.8 mM glucose KRB buffer with arginine (Sigma, A5131, 10 mM) and with or without Gal3 (250 ng/ml). Insulin concentrations in the supernatant were determined.

## GLP-1-stimulated insulin secretion assays

Mouse islets were washed with fresh 2.8 mM glucose KRB buffer. After fasting for 60 min with 2.8 mM glucose KRB buffer, cells were incubated for 60 min in fresh 2.8 mM or 16.8 mM glucose KRB buffer with or without GLP-1 (100 nM) and Gal3 (250 ng/ml). Insulin concentrations in the supernatant were determined.

## Islet perfusion assay

Primary murine islets were cultured overnight post-isolation before the perfusion assays. For each assay, perfusion chambers were loaded with 150 islet equivalents [IEQs] (approximately 100 size-matched islets) islets. Islets were first perfused with KRB buffer containing 2.8 mM glucose for 30 min, followed by perfusion with KRB buffers containing 2.8 mM glucose, 16.8 mM glucose and 2.8 mM glucose for 25, 30 and 20 min, respectively, under a 1 ml/min flow rate. Fractions were collected every minute. Islets were collected from the perfusion chamber after the assay to obtain the total protein content via the BCA protein assay. Insulin secretion at each time point was quantified using an HTRF insulin assay kit (Cisbio, 62INSPEC) according to the manufacturer's instructions, and normalized to the total protein content.

## Coculture experiments

Peritoneal macrophages were isolated from WT and Gal3$^{-/-}$ mice. Then, MIN6 cells ($8 \times 10^4$) with or without macrophages ($1 \times 10^5$) were seeded in 24-well plates. The medium was changed to remove unattached cells. GSIS was performed after 48 h. After GSIS experiments, the conditioned media was harvested to measure the insulin and Gal3 concentrations, and cells were harvested to measure the mRNA expression of *Lgals3* or other inflammatory factors.

For staining, MIN6 cells and macrophages were seeded on glass slides in 24-well plates. After harvesting conditioned media, the cells were fixed with 4% paraformaldehyde, and then permeated with 0.1% Triton X-100. Anti-insulin (rat; R&D, MAB1417, 1:200) and anti-CD11c (rabbit; CST, 97585, 1:100) antibodies were used as primary antibodies. Fluorescence images were acquired with a Zeiss LSM 880 microscope.

## Gal3 tracing experiments

Peritoneal macrophages were isolated from Gal3$^{-/-}$ mice (12 weeks of age). Then the macrophages were infected with adenovirus expressing GFP labeled Gal3 (Ad-Gal3) or adenovirus only expressing GFP (Ad-Con). After 36 h of infection, the macrophages were co-cultured with the MIN6 cells transfected with CACNG1 plasmid for 48 h. After 48 h of co-culture, the cells were performed for immunofluorescence staining. Anti-CACNG1 (1:200, LSBio, LS-C805513), anti-insulin (1:100, Abcam, ab7842) and anti-F4/80 (1:100, Abcam, ab6640, CI:A3-1) antibodies were used as primary antibodies, followed by incubation with secondary antibodies conjugated with Alexa546(Invitrogen, A11035, 1:200), Alexa647 (Invitrogen, A21450, 1:200) or Alexa594 (Invitrogen, A-11007, 1:200). The images were acquired by a Zeiss LSM 880 microscope.

## Glucose uptake in MIN6 cells

MIN6 cells were seeded in 24-well plates. After fasting with KRB buffer for 30 min, the cells were incubated at room temperature with 1 mM or 25 mM 2-deoxyglucose for 5 min. After adding stop buffer, the glucose uptake measurement was started following the instructions of the Glucose Uptake-Glo$^{TM}$ Assay Kit (Promega Corporation, USA, 017317).

## Cell viability detection

MIN6 cells were seeded in 96-well plates. After indication for 48 h, the cells were treated with Gal3 (125 ng/ml) for 6 h. Then we performed a cell viability experiment. The Cell Counting Kit-8 (CCK-8) assay (DOJINDO, Japan) was used according to the manufacturer's protocol. Ten microliters of CCK-8 was added to each well and cultured for an additional 1 h, and the absorbance at 450 nm was measured.

## Gal3 level detection

For plasma Gal3 levels, blood samples were drawn from the tail tip to measure Gal3 (Aviscera Bioscience, USA, SK00199-03). For islet Gal3 levels, primary islets were isolated from mice and then lysed in RIPA buffer to measure Gal3.

## ITTs, GTTs, and hyperglycemic clamp study

Male WT, Gal3$^{-/-}$ and Gal3$^{+/-}$ mice were fed a 60% HFD from 8 weeks of age for 8–16 weeks. For glucose tolerance tests (GTTs), the animals were intraperitoneally (i.p.) injected with glucose (1 g/kg) after 6 h of fasting, and blood was drawn to measure glucose levels at 0, 15, 30, 60, 90 and 120 min after glucose injection. Blood samples were collected from the tail tip at 0 and 15 min after glucose injection to measure insulin. For the intravenous glucose tolerance test, mice were injected glucose (1 g/kg) after overnight fasting, and blood was drawn to measure glucose at 0, 1, 5, 10, 30 and 60 min and to measure insulin at 0, 1, 5 and 10 min.

For *db/db* and Gal3$^{-/-}$ *db/db* male mice, IPGTTs and primary insulin secretion were performed at 8, 9, 12, and 14 weeks of age, and ITT was performed at 15 weeks of age. For IPGTTs and insulin secretion, mice

were fasted for 6 h and basal blood samples were taken, followed by i.p. injection with glucose (1 g/kg). Blood samples were drawn at 15, 30, 60, 90 and 120 min. For ITT, 0.4 U/kg insulin was subcutaneously injected after 6 h of fasting, and blood was drawn at 0, 15, 30, 60, 90, and 120 min after insulin injection.

Hyperglycemic clamps were performed as previously described in ref. 57. Briefly, the animals were fasted overnight and then anesthetized with pentobarbital sodium (50 mg/kg body weight, intraperitoneally) and placed on a heating pad at 37 °C. The anesthetics were reinforced every 60 min. The right jugular vein was catheterized for the infusion of glucose. After the operation, the animal rested for 30 min to lessen the stimulation. Then glucose was rapidly injected at a dose of 0.25 g/kg body weight in 1 min, followed by a continuous infusion of glucose (10%). Blood samples were collected initially (0 min) and then every 5 or 10 min throughout the test. The plasma glucose concentration was monitored instantaneously to adjust the infusion rate of glucose and to maintain hyperglycemia at nearly $14 \pm 0.5$ mmol/l. Blood samples were taken in heparinized tubes at 0, 1, 5, 10, 30, 60, 90 and 120 min and the plasma was stored at −20 °C for insulin analysis.

## Clodronate treatment

C57BL6/J mice were fed a 60% HFD (Diet Research) for 13 weeks from 8 weeks of age and were treated with clodronate as previously described[6].

## Gal3 inhibitor treatment in *db/db* mice

Seven- to eight-week-old *db/db* mice were divided into two groups according to body weight, fasting blood glucose, fasting insulin levels and insulin secretion before the treatment. Mice were treated with the Gal3 inhibitor GB1107 (MCE, HY-114409, 10 mg/kg per day[32]) or vehicle (0.1% Tween 80 distilled water) by oral gavage. IPGTT (1 g/kg) was performed after 2 weeks of treatment, and ITT (0.4 U/kg) was performed after 3 weeks of treatment. Then, the islets were isolated for GSIS.

## Gal3 antibody treatment in *db/db* mice

Five-week-old *db/db* mice were intravenously injected with Gal3 antibodies (Abcam, ab2785, 200 µg/kg) every 3 days for a total of 5 consecutive injections. Additional injections were administered at the 13th and 14th weeks. The control group mice were injected with normal IgG (200 µg/kg). IPGTT (1 g/kg) was performed at 15 weeks. Then, the islets were isolated for GSIS.

## Quantitative real-time PCR

Total RNA was extracted from frozen pancreas or cells using TRIzol reagent (Invitrogen, USA). cDNA was synthesized and quantitative real-time PCR was performed as previously described in ref. 55. First-strand cDNA was synthesized using a High-Capacity cDNA Reverse Transcription Kit (Applied Biosystems, #4368813) with random primers. qPCR was carried out in a 20 µl reaction system using SYBR Green qPCR Master Mix (Bimake, B21203) on a QuantStudio 3 Real-Time PCR System (Applied Biosystems). Relative gene expression was calculated as the mRNA level normalized to that of a standard housekeeping gene (*36B4*) using the delta-delta Ct (ΔΔ Ct) method. The specificity of the PCR amplification was verified by melting curve analysis of the final products using QuantStudio software. Primer sequences for qPCR are listed in Supplementary Table 2, Supplementary Table 3, Supplementary Table 4 and Supplementary Table 5.

## Immunoblotting and immunoprecipitation

Cells were transfected with plasmid for 48 h. Then, the culture medium was removed, and the cells were washed with PBS. For a 100 mm dish, 800 µl lysis buffer was added, and adherent cells were scraped into a 1.5 ml Ep tube. Cell extracts were incubated in a rocker at 4 °C for

20–30 min. The cells were centrifuged at 12,000 g for 10 min at 4 °C, and then the supernatant was moved to a new Ep tube. Next, 50–100 µl of lysates was added to another tube as the input, and protein buffer was added. The input samples were heated to 98 °C for 5 min for denaturation. Ten microliters of bead-linked antibody of bait protein were added to the remaining lysates, and the mixture was incubated on a rocker at 4 °C overnight. When the incubation time was over, the supernatant lysis was removed, and 800 µl of lysis buffer was added to wash the beads on the rocker at 4 °C for 10 min. The beads were washed 3 times. The last remaining supernatant was removed, and 30–50 µl of 2× protein buffer was added. The IP samples were heated to 98 °C for 5 min for denaturation. Input and IP samples were separated by SDS–PAGE for Western blot analysis. Primary antibodies were as follows: Flag (MBL M185-3L, 1:5000); GFP (MBL 598, 1:1000).

### Immunofluorescence (IF) and immunohistochemistry (IHC)
The pancreas was fixed in 4% paraformaldehyde and embedded in paraffin. Then, 5 µm sections of the tissues were made for analysis. Some sections were subjected to H&E staining to observe the morphology of the islets. Other sections were incubated overnight at 4 °C for IF staining with primary antibodies as follows: CD11c (1:100, CST, 97585 S, D1V9Y), Gal3 (1:200, Abcam, ab2785, A3A12), insulin (1:100, Abcam, ab7842; 1:200, R&D, MAB1417, 182410), glucagon (1:200, CST, 2760 S), CACNG1 (1:200, LSBio, LS-C805513), F4/80 (1:100, Abcam, ab6640, CI:A3-1) followed by incubation with secondary antibodies conjugated with Alexa488 (Invitrogen, A11029, 1:200), Alexa647 (Invitrogen, A21247, A21450, 1:200), Alexa594 (Invitrogen, A11032, 1:200), Alexa546 (Invitrogen, A11035, 1:200). The nuclei were stained with DAPI (Invitrogen, P36935). The images were acquired by a Zeiss LSM 880 microscope. The relative pancreatic area of β cells was calculated by dividing the insulin-positive area by the total islet area measured using ImageJ software. The β-cell mass was calculated by dividing the insulin-positive area by the total pancreatic area using ImageJ software and multiplied by the pancreatic weight.

For MIN6 cell staining, CACNG1-Flag was expressed in MIN6 cells for 48 h. Then, the cells were treated with Gal3 (500 ng/ml, 1 h) with or without CBP (1 µg/ml, 1 h). Anti-CACNG1 (rabbit; LSBio, LS-C805513, 1:200) and anti-Gal3 (mouse; Abcam, ab2785, 1:200, A3A12) antibodies were used as primary antibodies. The cytomembrane was stained with Alexa Fluor™ 594 phalloidin (Invitrogen, USA, A12381, 1:500). Fluorescence images were acquired with a Zeiss LSM 880 microscope.

### Proximity ligation assay (PLA)
PLA was performed according to the manufacturer's protocol (Sigma, DUO92101-1KT). CACNG1-Flag was expressed in MIN6 cells for 48 h. Then, the cells were treated with Gal3 (500 ng/ml) for 1 h. Anti-CACNG1 (rabbit; LSBio, LS-C805513, 1:200) and anti-Gal3 (mouse; Abcam, ab2785, 1:200, A3A12) antibodies were used as primary antibodies. Fluorescence images were acquired with a Zeiss LSM 880 microscope.

### Bio-layer interferometry (BLI) assay
The extracellular domains of mouse CACNG1 (30-109 AA, including two glycosylation sites) protein with His tags were expressed in FreeStyle 293F cells and purified. Mouse Gal3 proteins were purchased from R&D. Then, CACNG1 was captured by Ni-NTA (NTA) biosensors (Sartorius, 18-5101). The equilibrium dissociation constant ($K_D$) of CACNG1 with Gal3 in concentration gradient was measured by a BLI system with the Octet platform of ForteBio at 30 °C.

### Calcium imaging in MIN6 cells and mouse islets
A confocal microscope (Zeiss LSM 710) with a 40×, 1.3 NA oil-immersion objective was used for imaging. Frames of 512 × 512 pixels or 1024 × 1024 pixels were taken at a rate of 1 s/frame. The total number of frames and the length of imaging were dependent upon the experimental design.

Calcium dyes were obtained from Thermo Fisher and dissolved in DMSO at a stock concentration of 1 mM. MIN6 cells were incubated for 15 min at 37 °C with Fluo-4 AM (1 µM) and fura-2 AM (2 µM) in KRHB solution with 2.8 mM glucose, with or without 16.8 mM glucose stimulation. MIN6 cells were infected with a genetically encoded calcium indicator, GCaMP6f (No.67564, Addgene) expression adenovirus. After 36 h of expression, MIN6 cells efficiently expressed GCaMP6f in the cytosol for calcium imaging. Mouse islets specifically expressing GCaMP6f in β cells were obtained by primary dissection and were culture in 35 mm glass bottom dishes (Cellvis, D35-20-1.5-N, USA). The fluorescence intensity ratio of Fluo-4 or GCaMP6f was plotted against the dynamics of cytosolic calcium. The intracellular $Ca^{2+}$ concentration ($[Ca^{2+}]i$) was measured by confocal microscopy using fura-2, a dual-wavelength ratiometric fluorescent calcium indicator. Fura-2 was excited by a laser at wavelengths of 340 and 380 nm, $[Ca^{2+}]i$ was calculated from the ratio of the fluorescence intensity at 340 nm (F340) to that at 380 nm (F380). The digital image data were processed and analyzed by Zeiss Zen Blue software and ImageJ.

### Calculation of the frequency and peak value of the calcium curves in β cell of islets
Individual β-cell regions were circled, and the fluorescence intensity from the 4 regions of interest (ROIs) of islet β cells with and without Gal3 treatment was calculated. The baseline fluorescence ($F_0$) was calculated as the average fluorescence intensity from 0 to 60 s for ROIs. The change in fluorescence for graphical presentation was calculated from the following expression: $F/F_0$. Based on the trace of $F/F_0$, the frequency of calcium oscillations was calculated by the number of calcium spikes occurring per minute. The peak value of the calcium transient was the value of the highest point of the calcium transient.

### Patch-clamp recording in single MIN6 cell
Patch-clamp recording experiments in single MIN6 cell were performed in the perforated patch-clamp configuration described in previous studies[58–60] to maintain intact cellular metabolism of β cells. Patch pipettes were pulled from borosilicate glass capillaries with a resistance of 2–5 MΩ when filled with internal solution. Membrane currents and action potentials were recorded with an EPC-9 patch-clamp amplifier and PULSE software (HEKA Elektronik, Germany). For Gal3 treatment experiments, MIN6 cells were dialyzed with a solution containing 80 ng/ml Gal3 for at least 2 min. The standard extracellular solution contained 118 mM NaCl, 5.6 mM KCl, 5 mM HEPES, 20 mM TEA-Cl, 1.2 mM KCl, 2.5 mM $CaCl_2$, and 5 mM D-glucose. The intracellular solution consisted of 152 mmol/l $CsCH_3SO_3$, 10 mmol/l CsCl, 10 mmol/l KCl, 1 mmol/l $MgCl_2$ and 5 mmol/l HEPES, with the pH adjusted to 7.35 using CsOH.

### Quantification and statistical analysis
The results are presented as the mean ± standard error of the mean (SEM) with the numbers of experiments or mice indicated in the figure legends. To assess whether the means of two groups were significantly different from each other, unpaired two-sided Student's $t$-test was used for statistical analyses using Prism 6 software (GraphPad software 6.0; Prism). P values of 0.05 or less were considered statistically significant. Degrees of significance are indicated in the figure legends. No methods were used to determine whether the data met the assumptions of the statistical approach.

### Reporting summary
Further information on research design is available in the Nature Portfolio Reporting Summary linked to this article.

## Data availability
The human protein−protein interactome data used in this study are available in the BioPlex 3.0 (https://bioplex.hms.harvard.edu/). All data

are available within the Article, Supplementary Information or Source Data file. Any additional information is available upon request to the corresponding author (Pingping Li, lipp@imm.ac.cn). Source data are provided with this paper.

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

## Acknowledgements

We thank Dr. Tao Xu from the Institute of Biophysics, Chinese Academy of Sciences, for providing MIN6 cells. And we thank Dr. Jingxiang Wu in the Institute of Materia Medica, Chinese Academy of Medical Sciences and Peking Union Medical College, for providing FreeStyle 293F cells. This study was supported by research grants from the National Natural Science Foundation China (82104263 to Q.J., 81622010 to P.L., 82104259 to Q.Z., and 82304591 to Y.W.), the National Key R&D Program of China (2017YFA0205400), the Chinese Academy of Medical Sciences (CAMS) Central Public-Interest Scientific Institution Basal Research Fund (2017RC31009 and 2018PT35004), the CAMS Innovation Fund for Medical Sciences (2021-I2M-1-026 and 2021-I2M-1-016), the Beijing Outstanding Young Scientist Program (BJJWZYJH01201910023028 to P.L.), and the Special Research Fund for Central Universities, Peking Union Medical College (3332021041 to Q.Z., 3332022047 to Y.W.).

## Author contributions

Q.J. performed most of the experiments and analyzed the data; Q.Z., X.W., X.P., and L.C. performed the calcium signaling experiments; Y.C., C.M. and X.L. performed the Western blotting experiments; R.W. and Z.-X.M. performed the islet perfusion assay; S.H., L.K. and Y.W. assisted with GTTs and ITTs; Y.C. and S.H. assisted with collecting tissues and gene expression measurements; S.W. provided the human islets; P.L. conceived the project and directed the research. Q.J., Q.Z., B.C., and P.L. wrote the manuscript.

## Competing interests

The authors declare no competing interests.
