## [Peer Review File · Nature Communications]

Galectin-3 Impairs Calcium Transients and β -Cell FunctionEditorial Note: This manuscript has been previously reviewed at another journal that is not operating a transparent peer review scheme. This document only contains reviewer comments and rebuttal letters for versions considered at *Nature Communications*.

REVIEWER COMMENTS

Reviewer #1 (Remarks to the Author):

The manuscript by Qian Jiang et al. has been improved with the revisions, which have addressed most of my previous concerns. The manuscript significantly improves the mechanistic understanding of how inflammation causes islet dysfunction via Gal3-mediated inhibition of beta cell calcium channels.

Reviewer #2 (Remarks to the Author):

The authors are greatly appreciated for their diligent efforts in addressing my questions, which have notably improved the manuscript. However, the main concern that remains at the current version pertains to the absence of direct evidence, either in vitro or in vivo, demonstrating the interaction between macrophage released Gal3 and its binding to Cav gamma 1 on beta cells, subsequently inhibiting insulin secretion. In vitro, it would be more solid if the authors used a Gal3-tracing approach to elucidate the precise process through which labeled Gal3, originating from macrophages, interacts with beta cells under a cocultured condition. For in vivo, while the deletion of macrophages through clodronate treatment showed promising results, it remains somewhat indirect. The experiments involving the injection of Gal3 antibodies or pharmacological Gal3 inhibitors would provide a more convincing validation.

Some minor points:

Figures 1i and j present data on glucose-stimulated insulin secretion (GSIS) in HFD mice and db/db mice. However, considering the many existing literature on this topic, it is unclear what specific points the authors aim to convey to readers through these figures.

The statement made in line 54, "People with maximum or near maximum insulin resistance have lost more than 80% of their beta-cell function," should be substantiated with relevant references to support this statement.

Reviewer #3 (Remarks to the Author):

1. In this report, the authors used β -cell lines, diabetic and pre-diabetic knockout (KO) mouse models, and primary islets from mice and humans to investigate the role of galectin-3 (Gal3) in β -cell function and insulin secretion. They describe the role of Gal3 with emphasis on its proinflammatory and pro-apoptotic actions. The authors conclude that Gal3 acutely reduces glucose-stimulated insulin secretion (GSIS) by binding to calcium voltage-gated channel auxiliary subunit gamma 1 (CACNG1) and inhibiting calcium influx.

This a potentially important study as it identifies macrophage-derived Gal3 as a leading actor in the pathogenesis of type 2 diabetes, also based on an older report from the same group showing that this lectin directly enhances macrophage chemotaxis and induces insulin resistance in hepatocytes, adipocytes and myocytes (Cell. 2016;167:973-984).

However, the present study has several weaknesses and inconsistencies that limit its validity. 1. The authors reduce Gal3 complexity to an inflammatory molecule; in the abstract, Gal-3 is defined an “inflammatory cytokine” and throughout the manuscript it is implied that only macrophages express Gal3. However, from a biological point of view, Gal3 is a carbohydrate binding protein that: - cannot be defined a cytokine, as cytokines are peptides that cannot cross the lipid bilayer of cells to enter the cytoplasm and exert their effect through interaction with specific receptors. Gal-3 can penetrate lipid bilayer by interacting directly with membrane lipids to form vesicles with diverse functions (Biochem Biophys Res Commun. 2005;338:1031-1036; Nat Rev Mol Cell Biol. 2014;15:430-431). Gal3 enters and exits the cells without the need for signal proteins. - cannot be defined an inflammatory molecule, as it has both pro- and anti-inflammatory actions, depending on multiple factors, such as type of inflammatory setting and target cell/tissue. - is widely expressed in mammalian cells.

References regarding the structure and function of this lectin are rather outdated.

R: We thank the reviewer for pointing out the mistake and appreciate the comments on the definition and description of Gal3. We completely agree with you and have corrected the statement about Gal3 and updated the references regarding the structure and function of this lectin. (Line 2 Page 1; Line 28 Page 2; Lines 82-85 Page 5; and Line 394 Page 16)

2. The authors conclude with emphasis that they “demonstrated that Gal3 is a key inflammatory factor” and that data “demonstrate a role for Gal3 in inflammation-induced cell dysfunction”. These claims are not supported by the data, as no changes in inflammatory genes (Tnf- α , Il-1 β , Ccl2, and Il6) were observed in Gal3 KO mice or following pharmacological inhibition of Gal3. Only Adgre1 (F4/80) and Itgax (CD11c) were reduced in KO mice or mice treated with a Gal3 inhibitor, suggesting a reduced number of macrophages with a more inflammatory phenotype (i.e., the same levels of inflammatory cytokine expression despite a reduction in the macrophage marker Adgre1-F4/80). In addition:

- Regarding macrophage polarization, upregulation of Gal-3 is a feature of the alternative macrophage (M2) phenotype, and release of Gal3 by alternatively activated macrophages sustains the M2 phenotype (J Immunol. 2008; 180:2650–2658). In addition, Gal3 expression suppresses polarization toward a proinflammatory phenotype (Arterioscler Thromb Vasc Biol. 2020;40:1491-1509). Therefore, macrophage polarization should be investigated by evaluating markers of M2 macrophage polarization (Arginase-I, IL-10, TGF- β and other anti-inflammatory cytokines). - Regarding the role of Gal3 in β -cell dysfunction, in the Introduction section it is stated that “Gal3 was found to trigger β -cell apoptosis and islet inflammation in T2DM mice”. There are conflicting data in the literature on this issue. Previous studies showed that, in pancreatic islets exposed to IL-1 β , Gal3 is the most upregulated protein and that its overexpression protects the β -cells from the cytotoxic effect of IL-1 β (Biochem and Biophys Res Comm. 2006; 344:406–415). In addition, Gal3 ablation was associated with increased inflammation in pancreatic islets from mice fed an HFD (Diabetes. 2013;62:1932–1944), and transgenic overexpression of Gal3 in pancreatic β -cells was

found to attenuate hyperglycemia in mice (Front Pharmacol. 2021;12:714683. Ref 50 of the manuscript). The authors should discuss their findings and compare them with all previous literature on the subject (not only the literature in agreement with the data presented in the manuscript).

R: We thank the reviewer for pointing out this issue. As the reviewer said the Gal3 KO mice were characterized by no changes in inflammatory genes (*Tnfa*, *Il-1 β* , *Ccl2*, and *Il6*) and a slight reduction in macrophage markers. We measured inflammatory genes and anti-inflammatory cytokines in islets from HFD-fed WT and Gal3^{+/-} mice. For inflammatory genes, the results were similar to those of the pancreas from Gal3 KO mice or following pharmacological inhibition of Gal3. In these results, *Adgre1* (F4/80), *Itgax* (CD11c) and *Tnfa* levels were reduced, while *Il-1 β* , *Ccl2* and *Il-6* levels were still not changed (Fig. 8a). The level of the anti-inflammatory cytokine *Tgfb1* was increased, while others (including *Argase1*, *Il10*, and *Mrc1*) were not changed (Fig. 8b). These results indicate that Gal3 KO slightly improved inflammation. So it is not accurate to say that we “demonstrated that Gal3 is a key inflammatory factor”. We have now rephrased the statement to Gal3 is an important factor impairing β -cell function. It is known that Gal3 has both proinflammatory and anti-inflammatory effects. Gal3 is mainly produced and secreted by monocytes/macrophages¹⁸, but it is also expressed in the cytoplasm, nucleus or cell surface of other cell types¹⁹⁻²³. Extracellular and intracellular Gal3 usually have opposite roles. Mostly, extracellular Gal3 plays a proinflammatory role²⁴⁻²⁶.

Fig. 8 Relative mRNA expression of proinflammatory (a) and anti-inflammatory (b) cytokines in islets from WT and Gal3^{+/-} HFD-fed mice. Two-tailed Student's t test was used, and statistical data are expressed as the mean \pm SEM. n = 4-5 mice (male, HFD fed 13 weeks from 8 weeks old). *p < 0.05.

C: There are still issues related to biased interpretation and data consistency.

As shown in Fig.8 of the rebuttal, the primary discrepancy in comparison to the entire pancreas lies in the reduction of TNF-alpha in the islets, which aligns to the reduction of *Adgre1* and *Itgax*. Notably, the other inflammatory cytokines remain unchanged, as observed in the whole pancreas. Consequently, it is evident that the fluctuations in TNF-alpha within the islets are not linked to the diminished levels of markers (*Adgre1* to *Itgax*) of the monocyte-macrophage lineage (monocytes, macrophages, dendritic cells), nor they are related to the decreased expression of Gal3 in these cells. The question arises: why does Gal3 appear to regulate TNF-alpha production differently in islet macrophages compared to non-islet macrophages, while not impacting other inflammatory cytokines? On the whole, these data do little to support the conclusion that “Gal3 KO slightly ameliorates inflammation”. Moreover, they raise concerns due to their weak biological plausibility. Another data point inconsistent with Gal3 biology is the increase in Arginase1 (not Argase1) and TGF- β 1 in the islets of Gal-3 ^{+/-} mice. This is particularly perplexing, given the well-established understanding that Gal3 plays a crucial role in alternative macrophage activation and tissue fibrosis.

R: There are conflicting data in the literature on the role of Gal3 in β -cell dysfunction. We have discussed this issue in the manuscript. In pancreatic islets exposed to IL-1 β , Gal3 is the most upregulated protein, and its overexpression protects β cells from the cytotoxic effect of IL-1 β ²³. Transgenic overexpression of Gal3 in pancreatic β cells was found to protect β cells from apoptosis

in T1DM mice²⁷. In these two reports, the increased Gal3 was overexpressed in β cells, namely intracellular Gal3 of β cells. In our studies, extracellular Gal3 impaired β -cell function.

C: The animal model used by the Authors is a whole-body KO for Gal3. This approach weakens the Authors' assertion that it is extracellular Gal3 causing damage to β cells. As discussed by the authors, Gal3 is also produced and secreted by β cells. To bolster their argument that the noted effects are mediated by extracellular Gal3 of hematopoietic origin, rather than β cell-derived, the Authors should incorporate β cell-specific conditional knockout and/or transgenic overexpression of Gal3 in pancreatic β cells.

R: This is consistent with another study that showed that membrane-bound or extracellular Gal3 induces T-cell apoptosis²⁸, whereas cytoplasmic Gal3 expression protects T cells against apoptosis²⁹. Pejnovic *et al.*³⁰ showed that Gal3 ablation was associated with increased inflammation in pancreatic islets from mice fed a HFD. However, in their studies, the body weight of WT mice was less than 30 g after 18 weeks of 60% HFD feeding, and their Gal3 KO mice gained approximately 5 g of body weight, which is rather inexplicable since long-term 60% HFD feeding significantly increase the weight of male WT mice in most reports. In our study, the body weight of WT mice reached approximately 38 g even after 12 weeks of HFD feeding, and the Gal3 KO mice had comparable body weights. The inflammation of pancreatic islets is related to the body weight of HFD-fed mice to some extent. Therefore, in the report of Pejnovic *et al.*³⁰, inflammation of pancreatic islets was increased in Gal3 KO mice fed a HFD.

3. The authors report that “Gal3 deficiency led to more glucose tolerance combined with an increase in blood insulin secretion upon glucose stimulation, with no change in weight gain in mice fed an HFD”. This is also at odds (in whole or in part) with most of the previous literature (Diabetes. 2013;62:1932–1944; PLoS One. 2013;8:e57915; Endocrinology. 2015;156,147–156) and the reasons for these discrepancies should be discussed.

R: With regard to metabolism, these earlier studies provide contradictory results without clear conclusions in common among them. For example, Pejnovic *et al.*³⁰ showed that Gal3 KO mice have increased body weight on a HFD compared with WT mice. However, in their studies, the WT mice were not obese on a HFD, with body weights less than 30 g after 18 weeks of 60% HFD feeding, while their Gal3 KO mice gained approximately 5 g of body weight, and this is rather inexplicable, since long-term 60% HFD feeding causes weight gain in male WT mice. In our experience, male C57BL/6 mice always gain significant weight after feeding a 60% HFD for 18 weeks, usually reaching body weights between 45–50 g. Therefore, even the 35 g Gal3 KO mice reported by Pejnovic *et al.* would be considered very lean compared with most reports. Thus, the “obese” phenotype of their Gal3 KO mice is only in comparison to the lean WT mice, which failed to gain weight while consuming a HFD. Pang *et al.*³¹ studied a different strain of Gal3 KO mice and reported a modest degree of obesity in the KOs compared to WT mice. However, after 12 weeks of 60% HFD feeding in C57BL/6 mice, the weight gain in both groups was also significantly less than we usually see, with final body weights of approximately 38 g compared to 45–50 g. These investigators reported that the Gal3 KOs displayed mild hyperglycemia and glucose intolerance, with no change in insulin sensitivity. They also found that systemic antibiotic treatment, normalized the blood glucose levels, with no effect on glucose intolerance in the Gal3 KOs. Based on this information, they attributed their phenotype, at least in part, to unspecified changes in the gut microbiome. Baek *et al.*³² used the same strain of Gal3 KO mice as we used and reported that body weight was reduced in the Gal3 KO mice on a HFD, contrast to the first two papers mentioned above. As our studies showed, this was related to reducing inflammation and improving metabolism. There are another two papers using female Gal3 KO mice, that researched atherosclerosis or NASH^{33,34}. Both papers showed that with HFD feeding, the WT and Gal3 KO mice gained the same amount of body weight and were most comparable to the current research. Additionally, the Gal3 KOs exhibited improved glucose intolerance, consistent with our results. Although we cannot reconcile all of the conflicting data from the above mentioned reports, we

recognize that breeding and housing changes, technical differences, and microbiome differences could all contribute. However, we believe that the results reported in the current studies are quite comprehensive with multiple different measures indicating metabolic improvement in Gal3 KO. This concept is supported by our findings of metabolic deterioration with Gal3 treatment and metabolic benefit with administration of a Gal3 inhibitor.

C: In their rebuttal, the Authors present a series of contradictory statements. For your convenience, I have underlined the inconsistent statements in the preceding text, and I will now provide a brief summary. Initially, the Authors indicate that, according to their study, WT and Gal3KO mice on a HFD exhibited comparable body weights, aligning with the data showed in Fig. 1b (approximately 36 g for Gal3 KO and 38 g for WT). Furthermore, they assert a correlation between inflammation of pancreatic islets and body weight in HFD-fed mice. However, they later mention that the weight gain in both groups (WT and KO) after 12 weeks of HFD feeding reported by Pang et al., was significantly less (38 g) than they (i.e., the Authors) usually see (45-50 g). Usually, but not, evidently, in this study. In accordance with their data, the Authors cite Baek et al., who demonstrated that Gal3 KO mice on an HFD exhibit reduced body weight compared to WT HFD mice (which is at variance with the data showed in Fig. 1b), correlating with reduced inflammation and improved metabolism. Following this, they revert to stating, akin to two other studies, that Gal3 KO mice gained a comparable amount of body weight. This oscillation between arguments regarding weight gain and its connection to inflammation and the metabolic phenotype of Gal3 KO mice in their rebuttal does not effectively address the concerns initially raised in the first round review.

4. The authors report that “*db/db* Gal3 KO mice were much more insulin sensitive ... and glucose tolerant ..., with lower body weight”. Obesity in *db/db* mice is due to leptin deficiency and consequent hyperphagia. How do they explain the reduction in body weight? Did they measure food intake and show that Gal3 KO *db/db* mice were less hyperphagic?

R: Apart from insulin, we found that administration of Gal3 inhibited glucagon secretion in primary islets (Extended Data Fig. 2f in the revised manuscript). The pancreas of *db/db* mice was characterized by a higher α -cell content. Next, we tested the plasma glucagon levels in *db/db* Gal3 KO mice and found that glucagon levels were significantly higher in *db/db* Gal3 KO mice than in *db/db* mice (Fig. 9). It has been reported that glucagon reduces body weight by suppressing appetite and increasing fatty acid oxidation³⁵. Therefore, the increase in glucagon levels may explain the reduction in body weight in *db/db* Gal3 KO mice.

Fig. 9 Plasma glucagon levels in *db/db* and *db/db* GKO mice. Two-tailed Student's t test was used, and statistical data are expressed as the mean \pm SEM. $n = 6$ mice (male, 17 weeks old). * $p < 0.05$.

C: Elevated levels of glucagon are observed in all types of diabetes. Patients with type 2 diabetes experience both fasting and postprandial hyperglucagonemia, which stimulates hepatic glucose production and contributes to hyperglycemia. Consequently, the discovery of increased glucagon levels in *db/db* Gal3 KO mice contradicts the improved metabolic control observed in these mice.

Providing data on glutathione levels in *db/m* mice would help better interpret the data regarding this issue. Moreover, if the Authors claim that the rise in glucagon can account for the reduced body weight of *db/db* Gal3 KO mice, but not C57BL6 Gal3 KO mice fed a HFD, through the anorectic effects of this hormone, they must present data on decreased food intake.

5. The authors state that “*db/db* GKO mice had a feathered appearance”. What does it mean?

R: This was a writing error. We apologize for that. We have changed the phrase to “*db/db* GKO mice featured improved first-phase insulin and C-peptide secretion starting at 9 weeks of age (Fig. 3q and Extended Data Fig. 3l) compared to *db/db* mice.”

C: OK.

6. The authors report that, in *db/db* mice, hyperinsulinemia is noted by 10 days of age and frank hyperglycemia with glucose values of 10-15 mM by 8 wk of age. The age dependent increase in plasma Gal3 (Fig. 3, panel l) seems more a consequence than a contributing factor to diabetes development. Please discuss this issue.

R: In *db/db* mice, hyperglycemia is usually noted by 8 weeks of age. The plasma Gal3 levels increased significantly in *db/db* mice compared with *db/m* mice at 7 weeks of age (Fig. 1b in the original manuscript).

Fig. 1b in the original manuscript. Gal3 levels in the plasma of *db/m* and *db/db* mice (7 to 14 weeks old; n=3 mice). Two-tailed Student’s t test was used, and statistical data are expressed as the mean ± SEM. *p < 0.05; **p < 0.01.

C: Knowing that hyperglycemia typically occurs approximately 8 weeks after the onset of hyperinsulinemia in this strain, and that hyperglycemia is a result of β cells failing to produce sufficient insulin to counteract insulin resistance, it is reasonable to infer that the functional decline of beta cells by 7 weeks of age is already significantly advanced. Therefore, while the relationship between blood glucose and plasma Gal3 levels indicates an association and does not establish causality or its direction, considering the pathophysiology of type 2 diabetes, these data suggest that the increase in Gal3 likely represents a consequence rather than a contributing factor to the glucose metabolism derangement. In addition, the discovery that the blood glucose levels of *db/db* GKO mice plateaued from 9 weeks onward (refer to Figure 10 in the rebuttal) eliminates the possibility of Gal3 playing a causal role in promoting the development of diabetes.

7. The authors report apparently contrasting or unclear data regarding age and blood glucose levels of *db/db* mice. It is not clear what is the age of *db/db* mice in panel (n) of figure 3 and in panel (r) of Fig.3. Basal blood glucose levels (time 0) of *db/db* mice are much higher in panel (n) than in graphs of panel (o). Yet, at 9, 12, and 14 wks the *db/db* mice should have already developed frank diabetes.

R: As Fig. 3k (in the original manuscript) mentioned, in Fig. 3n and Fig. 3r (in the original manuscript), the ages of the *db/db* mice were 15 weeks and 16 weeks, respectively. As shown in Fig. 10, the basal blood glucose levels (time 0) of *db/db* mice gradually increased as the mice aged, with approximately 200 mg/dl at 9 weeks old and approximately 300 mg/dl at 15 weeks old. However, the blood glucose of *db/db* GKO mice stopped increasing from 9 weeks of age, and the 2 groups

showed significant differences at 15 weeks of age. The ITT experiment (Fig. 3n in the original manuscript) in *db/db* mice was performed at 15 weeks old, so the basal blood glucose levels were much higher in Fig. 3n (in the original manuscript) than in the graphs of Fig. 3o (at 9,12, and 14 weeks, in the original manuscript).

Fig. 10 Fasting blood glucose of *db/db* and *db/db* GKO mice. n= 5-7 mice (male, 8- 15 weeks old) ***P<0.001.

C: It is possible that the observed disparity is attributable to the utilization of distinct unit scales. However, it appears to me that there is a significant difference in the baseline blood glucose levels of *db/db* mice between panel n and panel o at the 14-week mark.

8. The authors present unclear data on insulin secretion, as in Fig 3, panel (p) and Extended Data Fig. 3, panel (l), it seems that first phase insulin secretion is impaired in both WT and GKO mice starting from 12 wk.

R: In Fig. 3p and Extended Data Fig. 3l, the first phase insulin secretion is impaired in *db/db* GKO mice starting from 12 weeks, but in *db/db* mice, it is from 9 weeks.

C: The authors are correct; we confused the animal experimental models (*db/db*, not HFD).

9. The authors do not provide convincing proof that Gal-3 from infiltrated macrophages plays a role in acute macrophage induced β -cell dysfunction. Insulin levels (production) are similar in islets derived from normal chow (NC) or *db/db* mice treated with vehicle and exposed to high glucose conditions (Fig 2, panels c and d). This does not support the conclusion that Gal3 from infiltrated macrophages impairs GSIS. Only exogenous Gal3 reduces GSIS in islets from both NC and *db/db* mice, although to a greater extent in islets from *db/db* mice than in islets from NC mice. This points in favor of some other factor responsible for the greater reduction of GSIS in the islets of *db/db* mice. The finding that Gal3 from macrophages of WT-NC mice does not affect GSIS in MIN6 cells (extended data Fig.2, panels I and J) supports this conclusion.

R: The GSIS experiment reflects the ability of β cells to secrete insulin when exposed to high-glucose conditions after low-glucose conditions. The insulin fold change instead of absolute insulin levels indicates the β -cell function ability. Many papers from human and animal studies have shown that during the early stage of diabetes^{11, 36-38}, pancreatic β cells tend to secrete more insulin in the basal state but show less response to high-glucose stimulation (change in insulin). Some papers even report GSIS results as the insulin fold change instead of insulin levels³⁹⁻⁴¹. Consistently, the islets from *db/db* mice had higher basal insulin levels compared to those of NC-fed mice (0.021 vs. 0.008 ng/ μ g Pro). Upon high-glucose stimulation, however, the insulin fold change increased approximately 20 times in islets derived from NC mice (Fig. 11a, blue bar), while in islets from *db/db* mice, the insulin levels increased only approximately 7 times, as shown in Fig. 11b (blue bar). Thus, our data support the conclusion that Gal3 from infiltrated macrophages impairs GSIS.

C: This is an elegant argument to support a role for macrophage-derived Gal3 in GSIS impairment despite the lack of differences in GSIS between islets derived from *db/db* and those derived from WT NC mice. Physiopathologically speaking, however, what matters is the absolute quantity of insulin produced following stimulation with glucose, which is the same between *db/db* and WT NC mice. The data rather suggest that Gal3 from infiltrated macrophages may increase basal insulin levels, presenting a completely different meaning and potential metabolic consequences. However, the fact that the addition of exogenous Gal3 in basal conditions does not modify insulin production in the islets of *db/db* (and also NC, see Fig. 2 panels d and c) does not support this explanation and argues in favor of the effect of some other inflammatory factor, aside from Gal3, on these changes. In short, the effect of macrophages on insulin production does not seem to be dependent on Gal3. The effect of macrophages on basal insulin production is also observed in MIN6 cells, but in the opposite way compared to what is seen in the islets. In fact, both WT and KO macrophages induce a reduction in insulin levels."

R: With regard to the results in macrophages from NC-fed mice and MIN6 cell coculture experiments, in the NC-fed state, macrophages are not inflamed and secrete less Gal3^{24,42,43}. This would explain why macrophages from NC-fed mice did not affect β - cell function. This is consistent with the in vivo finding that the function of islets from NC-fed mice was not reduced.

C: Gal3 levels in the conditioned medium from the co-culture system of MIN6 cells are much higher in WT-NC macrophages than in KO-NC macrophages (or the absence of macrophages, as shown in extended data Fig. 2, panel I). These data support a role for macrophages independent of Gal3 production. The statement that 'This is consistent with the in vivo finding that the function of islets from NC-fed mice was not reduced' does not make sense. It is unclear what 'not reduced' refers to, as we are discussing WT mice on a standard diet (i.e., controls)

R: Additionally, we depleted macrophages in HFD-fed mice by clodronate treatment and found that there was a marked decrease in Gal3 levels in the circulation, islets and supernatant of islets (Fig. 12d, e and f) in the clodronate treatment group, along with decent macrophage depletion in the islets (Fig. 12g). Importantly, clodronate treatment led to an increase in GSIS in primary islets, glucose tolerance and insulin secretion in HFD-fed mice, also supporting the idea that Gal3 from infiltrated macrophages impairs GSIS (see response to Question #1 of Reviewer #3).

C: Together with the previous data, this new set of data demonstrates that macrophages play a role in insulin resistance and the development of type 2 diabetes. A reduction in Gal3 levels is associated with these favorable metabolic effects. These new experiments confirm the association and, in light of previous studies, particularly those conducted on isolated islets and MIN6 cells, suggest that the effects of macrophages are independent of their production of Gal3 to some extent (see my comment above to this issue).

Fig. 11 GSIS with Gal3 treatment (250 ng/ml) for 6 h in primary islets. (a) NC mice, (b) *db/db* mice. Two-tailed Student's t test was used, and statistical data are expressed as the mean \pm SEM. n = 4-6 mice (male, NC: 12 weeks old, *db/db*: 17 weeks old). *p < 0.05; **p < 0.01.

10. The authors do not provide convincing proof that “KO (or pharmacological inhibition) of Gal3 was accompanied by a significant increase in β -cell mass”. In fact, data demonstrate that the percentage of the islet area (fractional area) occupied by β - cells is increased, not the mean β -cell area. As the size of the islets is greatly reduced in the KO (and pharmacological treated) mice (see panels a-f of Fig 4, and panels a-c of Extended Data Fig. 6), the mean islet area must be measured, and the mean β -cell area calculated by multiplying the fractional β -cell area by the mean islet area and dividing by 100.

R: We thank the reviewer for the comments on the statement about β -cell mass. Actually, the data we showed were the β -cell area expressed as the percentage of islets. We measured the area of β cells and the islet where they are located, and then the percentage of the islet area occupied by β cells was calculated by using the β -cell area dividing by the islet area and multiplying by 100. We have corrected “ β cell mass” to “ β cell area (% islet)”.

C: I apologize if the request was unclear. What the Authors are demonstrating is an increase in the average fractional area of β cells (i.e., the percentage of the islet occupied by beta cells). The islets of KO mice are smaller, so the data presented does not offer information on β cell mass per islet. It is possible that the β cell mass per islet could even be reduced in KOs if the percentage decrease in average islet size is greater than the fractional increase in β cells. Therefore, the data shown do not provide useful information regarding changes in β cell mass.

11. The authors show protein-protein interaction between Gal3 and CANG1, which represents the main novelty of the study. However, this issue deserves to be investigated in more physiological conditions than those of MIN6 cells transfected with CANG1 cmc plasmid in the presence of added GFP-tagged Gal3. As only exogenous Gal3 (i.e., added Gal3), but not endogenous Gal3, reduces GSIS in mice islets and MIN6 cells (Fig 2, panels c and d and extended data Fig.2, see panels I and J), coimmunoprecipitation of Gal3 and CACNG1 should be assessed in mice islets from WT and KO db/db mice and in MIN6 cells cocultured with intraperitoneal macrophages from Gal3 KO and WT mice. This to exclude that what is observed in experiments already performed is an artifact due to the addition of exogenous Gal3, as it actually happens for the reduction of GSIS.

R: We thank the reviewer for the comment “The authors show protein–protein interaction between Gal3 and CANG1, which represents the main novelty of the study.” We agree with the reviewer that this issue should be investigated fully. First, we showed endogenous Gal3 and CACNG1 colocalization by Co-IP in the MIN6 and macrophage coculture system and immunofluorescence staining in islets from *db/db* and HFD-fed mice using Gal3 and CACNG1 antibodies (please see Fig. 6c and d in response to Question #7 of Reviewer #1). Second, we proved that the interaction of Gal3 and CACNG1 is direct by the PLA assay and BLI assay (please see Figure 6e and f in response to Question #7 of Reviewer #1). Third, we found the binding sites in CACNG1 and proved that the binding occurs through glycosylation (please see Fig. 6a and b in response to Question #7 of Reviewer #1). Finally, we generated CACNG1 KO and β - cell-specific CACNG1 KO mice and showed that CACNG1 is critical for β -cell function and in vivo glucose tolerance (please see Fig. 3 in response to Question #2 of Reviewer #1 and Fig. 13d-g in response to Question #2 of Reviewer #3).

C: These new experiments demonstrate that CANG1 is critical for β cell function and glucose tolerance. Gal3 and CANG1 colocalize in islets from *db/db* and HFD-fed mice. One of the points to clarify, as originally requested by this Reviewer, is whether the interaction occurs physiologically or only in obese/diabetic animals; that is, does the interaction have pathophysiological relevance? Unfortunately, the new experiments did not include metabolically healthy WT mice. Gal3 functions as an endocytic adapter that co-clusters glycosylated cargos and membrane glycosphingolipids, leading to both membrane bending and clathrin-independent formation of the endocytic pit (PMID: 24837829). In addition, the formation of a cross-linked lattice between galectin-3 and cell surface glycoprotein receptors was shown to inhibit endocytosis of some receptors and regulate their availability for signaling. Therefore, it is not surprising that Gal3 and CACNG1 colocalize in islets

from mice, including db/db and HFD-fed mice, and pathomechanistic conclusions cannot be drawn from these findings.

RESPONSE TO REVIEWERS' COMMENTS

Reviewer #1 (Remarks to the Author):

The manuscript by Qian Jiang et al. has been improved with the revisions, which have addressed most of my previous concerns. The manuscript significantly improves the mechanistic understanding of how inflammation causes islet dysfunction via Gal3-mediated inhibition of beta cell calcium channels.

R: We thank Reviewer #1 for the encouraging comments that “The manuscript significantly improves the mechanistic understanding of how inflammation causes islet dysfunction via Gal3-mediated inhibition of β -cell calcium channels.”

Reviewer #2 (Remarks to the Author):

The authors are greatly appreciated for their diligent efforts in addressing my questions, which have notably improved the manuscript. However, the main concern that remains at the current version pertains to the absence of direct evidence, either *in vitro* or *in vivo*, demonstrating the interaction between macrophage released Gal3 and its binding to Cav gamma 1 on beta cells, subsequently inhibiting insulin secretion. *In vitro*, it would be more solid if the authors used a Gal3-tracing approach to elucidate the precise process through which labeled Gal3, originating from macrophages, interacts with beta cells under a cocultured condition. For *in vivo*, while the deletion of macrophages through clodronate treatment showed promising results, it remains somewhat indirect. The experiments involving the injection of Gal3 antibodies or pharmacological Gal3 inhibitors would provide a more convincing validation.

R: We thank Reviewer #2 for the compliment that “The authors are greatly appreciated for their diligent efforts in addressing my questions, which have notably improved the manuscript.” With regard to the concerns of Reviewer #2 about the interaction between macrophage-released Gal3 and Cav gamma 1 on β cells, we performed the Gal3-tracing experiment *in vitro*. Macrophages from Gal3 KO mice were isolated, followed by infection of the adenovirus expressing Gal3-GFP (Ad-Gal3) or control virus (**Fig.18a**). After 36 h of infection, the macrophages were co-cultured with the MIN6 cells transfected with CACNG1 plasmid for 48 h. After 48 h of co-culture, the cells were performed for immunofluorescence staining. The results showed that in macrophages and MIN6 cells co-culture system, GFP labeled Gal3 colocalized with CACNG1 on MIN6 cells when the Gal3 KO macrophages were infected with adenovirus expressing Gal3-GFP (Ad-Gal3) (**Fig.18a**). These results suggest that macrophage-released Gal3 could bind to CACNG1 on β cells.

Moreover, we generated macrophage Gal3 knockout mice with Cre loxp system (**Fig. 18b**). As **Fig. 18c** showed, compared to Gal3^{f/f} mice, the circulating Gal3 levels were approximately 65% and 50% lower in Gal3^{f/+}-lyz Cre and Gal3^{f/f}-lyz Cre mice on HFD, respectively, with comparable body weight among the groups (**Fig. 18d**). Similarly, the Gal3 levels were about 90% decreased in islets from Gal3^{f/f}-lyz Cre mice (**Fig. 18e**). Of note, macrophage-specific Gal3 depletion led to an improvement in glucose tolerance in HFD-fed mice and an increase of insulin secretion during IPGTT (**Fig. 18f** and **18g**). In line with the *in vivo* data, in primary islets from Gal3^{f/f}-lyz Cre mice on HFD, GSIS was higher compared to Gal3^{f/f} group (**Fig. 18h**). Together, these results suggested that the Gal3 in macrophages accounts for the impairment of β -cell function *in vivo*.

We did perform experiments with the Gal3 inhibitors and antibody in our original manuscript (**Fig 7a-e** in the 1st revised manuscript). For example, Gal3 inhibitors (GB1107, TD139, Cpd47) and Gal3 antibody reversed the effect of Gal3 on GSIS in MIN6 cells (**Fig. 7a-c** in the 1st revised manuscript). Additionally, Gal3 inhibitor GB1107 reversed the effect of Gal3 on the cytosolic calcium transient trace and peak values in MIN6 cells (**Fig. 7d, e** in the 1st revised manuscript) and improved glucose tolerance

with enhanced insulin secretion in *db/db* mice (Fig. 7g-l in the 1st revised manuscript). Additionally, we treated *db/db* mice with Gal3 antibody and found similar results as Gal3 inhibitor (Fig. 18i-l). Taken together, these data in Gal3 inhibitors and Gal3 antibody was consistent with the results in clodronate injected mice, demonstrating the role of Gal3 in macrophages on β -cell dysfunction.

Fig. 18 Macrophage-specific Gal3 knockout improved glucose intolerance and GSIS. a, Gal3-tracing approach, GFP labeled Gal3 derived from macrophages, interacted with CACNG1 in MIN6 cells under a co-cultured condition. Peritoneal macrophages were isolated from Gal3^{-/-} mice (12 weeks of age). Then the macrophages were infected with adenovirus expressing GFP labeled Gal3 (Ad-Gal3) or adenovirus only expressing GFP (Ad-Con). After 36 h of infection, the macrophages were co-cultured with the MIN6 cells transfected with CACNG1 plasmid for 48 h. After 48 h of co-culture, the cells were performed for immunofluorescence staining. Anti-CACNG1 (1:100, Novus, NBP2-76917), anti-insulin (1:100, Abcam, ab7842), and anti-F4/80 (1:100, Abcam, ab6640) antibodies were used as primary

antibodies, followed by incubation with secondary antibodies conjugated with Alexa546, Alexa647 or Alexa594 (Invitrogen, USA). The images were acquired by a Zeiss LSM 880 microscope. Scale bar, 20 μm and 5 μm . **b**, Schematic representation of conditional floxed and deleted *Lgals3* gene loci. **c-d**, Gal3 levels in the blood (**c**) and islets (**d**) from HFD-fed mice. **e-h**, Body weight (**e**), IPGTT (**f**), first-phase insulin secretion (**g**) and GSIS in primary islets (**h**) in Gal3^{f/f}, Gal3^{f/+}-lyzcre and Gal3^{f/f}-lyzcre HFD-fed mice, HFD:4 weeks from 8 weeks of age. **i**, Experimental scheme of *db/db* mice by Gal3 antibody injection. At the age of 5 weeks, *db/db* mice were intravenously injected with Gal3 antibodies every 3 days for a total of 5 consecutive injections. Additional injections were administered at the 13th and 14th weeks. **j-l**, IPGTT (15 weeks of age) (**j**), GSIS in primary islets (16 weeks of age) (**k**) and Blood Gal3 antibody levels (16 weeks of age) (**l**) in *db/db* mice. Two-tailed Student's t test was used, and statistical data were expressed as the mean \pm SEM. n = 5-7 per group (**c-h, j**), n = 4 per group (**k**), n = 9 per group (**l**). *p < 0.05; **p < 0.01; ***p < 0.001. #p < 0.05; ##p < 0.01. * Gal3^{f/f}-lyzcre vs Gal3^{f/f} mice (**f**), # Gal3^{f/+}-lyzcre vs Gal3^{f/f} mice (**f**).

Some minor points:

Figures 1i and j present data on glucose-stimulated insulin secretion (GSIS) in HFD mice and *db/db* mice. However, considering the many existing literature on this topic, it is unclear what specific points the authors aim to convey to readers through these figures.

R: In **Fig. 1g** and **h** (original and revised manuscript), we showed the Gal3 levels were higher in primary islets from HFD-fed and *db/db* mice than those from NC-fed and *db/m* mice. **Fig. 1i** and **j** (original and revised manuscript) showed that GSIS was reduced in HFD-fed and *db/db* mice. So here we would like to express that β -cell dysfunction may be related to the increase of Gal3 levels in islets in animal models.

The statement made in line 54, "People with maximum or near maximum insulin resistance have lost more than 80% of their beta-cell function," should be substantiated with relevant references to support this statement.

R: The reference (ref. 3) has been added to substantiate the statement made in line 56 (in the 2nd revised manuscript), "People with maximum or near maximum insulin resistance have lost more than 80% of their beta-cell function".

Note: Initial comments from first round review are in **black**, and response to initial comments is in **blue**; comments (Reviewer #3) from second round review are in **red**, and response to these comments is in **green**.

Reviewer #3 (Remarks to the Author):

1. In this report, the authors used β -cell lines, diabetic and pre-diabetic knockout (KO) mouse models, and primary islets from mice and humans to investigate the role of galectin-3 (Gal3) in β -cell function and insulin secretion. They describe the role of Gal3 with emphasis on its proinflammatory and pro-apoptotic actions. The authors conclude that Gal3 acutely reduces glucose-stimulated insulin secretion (GSIS) by binding to calcium voltage-gated channel auxiliary subunit gamma 1 (CACNG1) and inhibiting calcium influx.

This a potentially important study as it identifies macrophage-derived Gal3 as a leading actor in the pathogenesis of type 2 diabetes, also based on an older report from the same group showing that this lectin directly enhances macrophage chemotaxis and induces insulin resistance in hepatocytes, adipocytes and myocytes (Cell. 2016;167:973-984).

However, the present study has several weaknesses and inconsistencies that limit its validity. 1. The authors reduce Gal3 complexity to an inflammatory molecule; in the abstract, Gal-3 is defined an

“inflammatory cytokine” and throughout the manuscript it is implied that only macrophages express Gal3. However, from a biological point of view, Gal3 is a carbohydrate binding protein that: - cannot be defined a cytokine, as cytokines are peptides that cannot cross the lipid bilayer of cells to enter the cytoplasm and exert their effect through interaction with specific receptors. Gal-3 can penetrate lipid bilayer by interacting directly with membrane lipids to form vesicles with diverse functions (Biochem Biophys Res Commun. 2005;338:1031-1036; Nat Rev Mol Cell Biol.

2014;15:430-431). Gal3 enters and exits the cells without the need for signal proteins.

- cannot be defined an inflammatory molecule, as it has both pro- and anti-inflammatory actions, depending on multiple factors, such as type of inflammatory setting and target cell/tissue.

- is widely expressed in mammalian cells.

References regarding the structure and function of this lectin are rather outdated.

R: We thank the reviewer for pointing out the mistake and appreciate the comments on the definition and description of Gal3. We completely agree with you and have corrected the statement about Gal3 and updated the references regarding the structure and function of this lectin. (Line 2 Page 1; Line 28 Page 2; Lines 82-85 Page 5; and Line 394 Page 16)

2. The authors conclude with emphasis that they “demonstrated that Gal3 is a key inflammatory factor” and that data “demonstrate a role for Gal3 in inflammation-induced cell dysfunction”. These claims are not supported by the data, as no changes in inflammatory genes (*Tnf- α* , *Il-1 β* , *Ccl2*, and *Il6*) were observed in Gal3 KO mice or following pharmacological inhibition of Gal3. Only *Adgre1* (F4/80) and *Itgax* (CD11c) were reduced in KO mice or mice treated with a Gal3 inhibitor, suggesting a reduced number of macrophages with a more inflammatory phenotype (i.e., the same levels of inflammatory cytokine expression despite a reduction in the macrophage marker *Adgre1*- F4/80). In addition:

- Regarding macrophage polarization, upregulation of Gal-3 is a feature of the alternative macrophage (M2) phenotype, and release of Gal3 by alternatively activated macrophages sustains the M2 phenotype (J Immunol. 2008; 180:2650–2658). In addition, Gal3 expression suppresses polarization toward a proinflammatory phenotype (Arterioscler Thromb Vasc Biol. 2020;40:1491- 1509). Therefore, macrophage polarization should be investigated by evaluating markers of M2 macrophage polarization (*Arginase-I*, *IL-10*, *TGF- β* and other anti-inflammatory cytokines). - Regarding the role of Gal3 in β -cell dysfunction, in the Introduction section it is stated that “Gal3 was found to trigger β -cell apoptosis and islet inflammation in T2DM mice”. There are conflicting data in the literature on this issue. Previous studies showed that, in pancreatic islets exposed to *IL-1 β* , Gal3 is the most upregulated protein and that its overexpression protects the β -cells from the cytotoxic effect of *IL-1 β* (Biochem and Biophys Res Comm. 2006; 344:406–415). In addition, Gal3 ablation was associated with increased inflammation in pancreatic islets from mice fed an HFD (Diabetes. 2013;62:1932–1944), and transgenic overexpression of Gal3 in pancreatic β -cells was found to attenuate hyperglycemia in mice (Front Pharmacol. 2021;12:714683. Ref 50 of the manuscript). The authors should discuss their findings and compare them with all previous literature on the subject (not only the literature in agreement with the data presented in the manuscript).

R: We thank the reviewer for pointing out this issue. As the reviewer said the Gal 3 KO mice were characterized by no changes in inflammatory genes (*Tnfa*, *Il-1 β* , *Ccl2*, and *Il6*) and a slight reduction in macrophage markers. We measured inflammatory genes and anti-inflammatory cytokines in islets from HFD-fed WT and Gal3^{+/-} mice. For inflammatory genes, the results were similar to those of the pancreas from Gal3 KO mice or following pharmacological inhibition of Gal3. In these results, *Adgre1* (F4/80), *Itgax* (CD11c) and *Tnfa* levels were reduced, while *Il-1 β* , *Ccl2* and *Il-6* levels were still not changed (Fig. 8a). The level of the anti-inflammatory cytokine *Tgfb1* was increased, while others (including *Argase1*, *Il10*, and *Mrc1*) were not changed (Fig. 8b). These results indicate that Gal3 KO slightly improved inflammation. So it is not accurate to say that we “demonstrated that Gal3 is a key inflammatory factor”. We have now rephrased the statement to Gal3 is an important factor impairing β -cell function. It is known that Gal3 has both proinflammatory and anti-inflammatory effects. Gal3 is mainly produced and secreted by monocytes/macrophages¹⁸, but it is also expressed

in the cytoplasm, nucleus or cell surface of other cell types¹⁹⁻²³. Extracellular and intracellular Gal3 usually have opposite roles. Mostly, extracellular Gal3 plays a proinflammatory role²⁴⁻²⁶.

Fig. 8 Relative mRNA expression of proinflammatory (a) and anti-inflammatory (b) cytokines in islets from WT and Gal3+/- HFD-fed mice. Two-tailed Student's t test was used, and statistical data are expressed as the mean \pm SEM. n = 4-5 mice (male, HFD fed 13 weeks from 8 weeks old). *p < 0.05.

C: There are still issues related to biased interpretation and data consistency.

As shown in Fig.8 of the rebuttal, the primary discrepancy in comparison to the entire pancreas lies in the reduction of TNF-alpha in the islets, which aligns to the reduction of Adgre1 and Itgax.

Notably, the other inflammatory cytokines remain unchanged, as observed in the whole pancreas. Consequently, it is evident that the fluctuations in TNF-alpha within the islets are not linked to the diminished levels of markers (Adgre1 to Itgax) of the monocyte-macrophage lineage (monocytes, macrophages, dendritic cells), nor they are related to the decreased expression of Gal3 in these cells. The question arises: why does Gal3 appear to regulate TNF-alpha production differently in islet macrophages compared to non-islet macrophages, while not impacting other inflammatory cytokines? On the whole, these data do little to support the conclusion that "Gal3 KO slightly ameliorates inflammation". Moreover, they raise concerns due to their weak biological plausibility. Another data point inconsistent with Gal3 biology is the increase in Arginase1 (not Argase1) and TGF- β 1 in the islets of Gal-3 +/- mice. This is particularly perplexing, given the well-established understanding that Gal3 plays a crucial role in alternative macrophage activation and tissue fibrosis.

R: In our manuscript, we found that TNF-alpha is decreased, along with less macrophage accumulation in the islets from Gal3 +/- HFD-fed mice compared to control mice. So we concluded that "Gal3 KO slightly ameliorates inflammation". The reviewer thinks these data do little to support the conclusion, now we removed this conclusion in the 2nd revised manuscript. (Page 11 in the 2nd revised manuscript).

The main finding in this manuscript is that Gal3 acutely impairs calcium transients (a few minutes) and β -cell function. And the mechanism is through the direct binding of Gal3 with CACNG1. The effect of Gal3 on inflammatory and anti-inflammatory gene expression would be the secondary effect since it needs several hours for gene expression. Thus, the effect of Gal3 on β -cell function is independent of inflammation. We would like to point out that the tissue we are measuring is islets from the HFD-fed mice, it's possible that different tissues and animal conditions (diet, age) sometimes give conflicting results. We removed the description about inflammation in the results section (Page 10 in the 2nd revised manuscript).

Thanks for pointing out the spelling mistake about Arginase, we corrected it

C: The animal model used by the Authors is a whole-body KO for Gal3. This approach weakens the Authors' assertion that it is extracellular Gal3 causing damage to β cells. As discussed by the authors, Gal3 is also produced and secreted by β cells. To bolster their argument that the noted effects are

mediated by extracellular Gal3 of hematopoietic origin, rather than β cell-derived, the Authors should incorporate β cell-specific conditional knockout and/or transgenic overexpression of Gal3 in pancreatic β cells.

R: Apart from the Gal3 KO mice, we studied other mice including the Gal3 f/f-LyzCre mice (Fig 18) and clodronate-treated mice on HFD (Fig 4 in the 2nd revised manuscript). The results in these mice consistently showed that Gal3 from macrophages impaired β -cell function. Furthermore, in the Gal3 tracing experiment, we found that macrophage secret Gal3 and it could bind to CACNG1 in MIN6 cells (Fig. 18). Along with the results that Gal3 supplement to the medium decreased GSIS in both β cell lines and primary islets from rodents and human subjects (Fig. 2 and 7r-t in the 2nd revised manuscript). With these results we believe that extracellular Gal3 from macrophage takes the effect on β -cell dysfunction. It's true that Gal3 is also expressed by β cells. However, the expression of Gal3 in β cells is very low compared to macrophages according to the open-source public single-cell RNA sequencing data (Fig. 19). In this study, we are trying to convince you that Gal3 from macrophage impair β -cell function, so we didn't design studies with β cell-specific conditional knockout and/or transgenic overexpression of Gal3 in pancreatic β cells.

Fig. 19 LGALS3 expression in islet cell subtypes. a, healthy human islets, b, diabetes human islets, c, normal mouse islets, d, HFD-fed mouse islets. Human LGALS3 expression in islet cell subtypes extracted from an integrated dataset of human single-cell RNA-seq data. The dataset from project E-MTAB-5061 contains 3514 cells from 6 healthy people and 4 diabetic patients. Mouse *Lgals3* expression in islet cell subtypes extracted from an integrated dataset of mouse single-cell RNA-seq project GSE162512. The dataset contains 68301 cells from 3 NC mice and 4 HFD-fed mice.

R: This is consistent with another study that showed that membrane-bound or extracellular Gal3 induces T-cell apoptosis²⁸, whereas cytoplasmic Gal3 expression protects T cells against apoptosis²⁹. Pejnovic *et al.*³⁰ showed that Gal3 ablation was associated with increased inflammation in pancreatic islets from mice fed a HFD. However, in their studies, the body weight of WT mice was less than 30 g after 18 weeks of 60% HFD feeding, and their Gal3 KO mice gained approximately 5 g of body weight, which is rather inexplicable since long-term 60% HFD feeding significantly increase the weight of male WT mice in most reports. In our study, the body weight of WT mice reached approximately 38 g even after 12 weeks of HFD feeding, and the Gal3 KO mice had comparable body weights. The inflammation of pancreatic islets is related to the body weight of HFD-fed mice to some extent. Therefore, in the report of Pejnovic *et al.*³⁰, inflammation of pancreatic islets was increased in Gal3 KO mice fed a HFD.

3. The authors report that “Gal3 deficiency led to more glucose tolerance combined with an increase in blood insulin secretion upon glucose stimulation, with no change in weight gain in mice fed an HFD”. This is also at odds (in whole or in part) with most of the previous literature (Diabetes. 2013;62:1932–1944; PLoS One. 2013;8:e57915; Endocrinology. 2015;156,147–156) and the reasons for these discrepancies should be discussed.

R: With regard to metabolism, these earlier studies provide contradictory results without clear conclusions in common among them. For example, Pejnovic *et al.*³⁰ showed that Gal3 KO mice have increased body weight on a HFD compared with WT mice. However, in their studies, the WT mice were not obese on a HFD, with body weights less than 30 g after 18 weeks of 60% HFD feeding, while their Gal3 KO mice gained approximately 5 g of body weight, and this is rather inexplicable, since long-term 60% HFD feeding causes weight gain in male WT mice. In our experience, male C57BL/6 mice always gain significant weight after feeding a 60% HFD for 18 weeks, usually reaching body weights between 45–50 g. Therefore, even the 35 g Gal3 KO mice reported by Pejnovic *et al.* would be considered very lean compared with most reports. Thus, the “obese” phenotype of their Gal3 KO mice is only in comparison to the lean WT mice, which failed to gain weight while consuming a HFD. Pang *et al.*³¹ studied a different strain of Gal3 KO mice and reported a modest degree of obesity in the KOs compared to WT mice. However, after 12 (spelling mistake, it should be 18) weeks of 60% HFD feeding in C57BL/6 mice, the weight gain in both groups was also significantly less than we usually see, with final body weights of approximately 38 g compared to 45–50 g. These investigators reported that the Gal3 KOs displayed mild hyperglycemia and glucose intolerance, with no change in insulin sensitivity. They also found that systemic antibiotic treatment, normalized the blood glucose levels, with no effect on glucose intolerance in the Gal3 KOs. Based on this information, they attributed their phenotype, at least in part, to unspecified changes in the gut microbiome. Baek *et al.*³² used the same strain of Gal3 KO mice as we used and reported that body weight was reduced in the Gal3 KO mice on a HFD, contrast to the first two papers mentioned above. As our studies showed, this was related to reducing inflammation and improving metabolism. There are another two papers using female Gal3 KO mice, that researched atherosclerosis or NASH^{33,34}. Both papers showed that with HFD feeding, the WT and Gal3 KO mice gained the same amount of body weight and were most comparable to the current research.

Additionally, the Gal3 KOs exhibited improved glucose intolerance, consistent with our results. Although we cannot reconcile all of the conflicting data from the above mentioned reports, we recognize that breeding and housing changes, technical differences, and microbiome differences could all contribute. However, we believe that the results reported in the current studies are quite comprehensive with multiple different measures indicating metabolic improvement in Gal3 KOs. This concept is supported by our findings of metabolic deterioration with Gal3 treatment and metabolic benefit with administration of a Gal3 inhibitor.

C: In their rebuttal, the Authors present a series of contradictory statements. For your convenience, I have underlined the inconsistent statements in the preceding text, and I will now provide a brief summary. Initially, the Authors indicate that, according to their study, WT and Gal3KO mice on a HFD exhibited comparable body weights, aligning with the data showed in Fig. 1b (approximately 36 g for Gal3 KO and 38 g for WT). Furthermore, they assert a correlation between inflammation of pancreatic islets and body weight in HFD-fed mice. However, they later mention that the weight gain in both groups (WT and KO) after 12 (it should be 18) weeks of HFD feeding reported by Pang *et al.*, was significantly less (38 g) than they (i.e., the Authors) usually see (45-50 g). Usually, but not, evidently, in this study. In accordance with their data, the Authors cite Baek *et al.*, who demonstrated that Gal3 KO mice on an HFD exhibit reduced body weight compared to WT HFD mice (which is at variance with the data showed in Fig. 1b), correlating with reduced inflammation and improved metabolism. Following this, they revert to stating, akin to two other studies, that Gal3 KO mice gained a comparable amount of body weight. This oscillation between arguments regarding weight gain and its connection to inflammation and the metabolic phenotype of Gal3 KO mice in their rebuttal does not effectively address the concerns initially raised in the first round review.

R: There is a spelling mistake, the mice were 18 weeks of HFD instead of 12 weeks (see the highlighted part). We are very sorry for the mistake and confusion.

4. The authors report that “db/db Gal3 KO mice were much more insulin sensitive ... and glucose tolerant, with lower body weight”. Obesity in db/db mice is due to leptin deficiency and consequent hyperphagia. How do they explain the reduction in body weight? Did they measure food

intake and show that Gal3 KO *db/db* mice were less hyperphagic?

R: Apart from insulin, we found that administration of Gal3 inhibited glucagon secretion in primary islets (Extended Data Fig. 2f in the revised manuscript). The pancreas of *db/db* mice was characterized by a higher α -cell content. Next, we tested the plasma glucagon levels in *db/db* Gal3 KO mice and found that glucagon levels were significantly higher in *db/db* Gal3 KO mice than in *db/db* mice (Fig. 9). It has been reported that glucagon reduces body weight by suppressing appetite and increasing fatty acid oxidation³⁵. Therefore, the increase in glucagon levels may explain the reduction in body weight in *db/db* Gal3 KO mice.

Fig. 9 Plasma glucagon levels in *db/db* and *db/db* GKO mice. Two-tailed Student's t test was used, and statistical data are expressed as the mean \pm SEM. $n = 6$ mice (male, 17 weeks old). * $p < 0.05$.

C: Elevated levels of glucagon are observed in all types of diabetes. Patients with type 2 diabetes experience both fasting and postprandial hyperglucagonemia, which stimulates hepatic glucose production and contributes to hyperglycemia. Consequently, the discovery of increased glucagon levels in *db/db* Gal3 KO mice contradicts the improved metabolic control observed in these mice. Providing data on glutathione levels in *db/m* mice would help better interpret the data regarding this issue. Moreover, if the Authors claim that the rise in glucagon can account for the reduced body weight of *db/db* Gal3 KO mice, but not C57BL6 Gal3 KO mice fed a HFD, through the anorectic effects of this hormone, they must present data on decreased food intake.

R: As suggested, we measured blood glutathione (GSH) levels in *db/m*, *db/db*, and *db/db*-Gal3 KO mice. The results showed that compared to *db/m* mice, the blood GSH levels in *db/db* mice were significantly reduced (Fig. 20a). After the knockout of Gal3, the blood GSH levels in *db/db* mice show a marked increase (Fig. 20a). GSH is known to enhance energy metabolism, so the reduced body weight in *db/db* GKO mice may be related to the increased GSH levels.

We could not get the food intake data in *db/db* GKO mice, since these mice didn't breed well and we are losing this line. Luckily, the experiments involving the injection of Gal3 antibodies into *db/db* mice showed a significant decrease in body weight, which is similar to finding in *db/db* GKO mice (Fig. 20b). The antibody-treated mice exhibited a significant reduction in food intake (Fig. 20c).

Fig. 20 Gal3 antibody treatment decreased body weight and food intake in *db/db* mice. **a**, Plasma glutathione (GSH) levels in *db/m*, *db/db* and *db/db* Gal3 KO mice (17 weeks old). **b-c**, Body weight (**b**) and food intake (**c**) of *db/db* mice. At the age of 5 weeks, *db/db* mice were intravenously injected with Gal3 antibodies (200 µg/kg) every 3 days for a total of 5 consecutive injections. Additional injections were administered at the 13th and 14th weeks. Two-tailed Student's t test was used, and statistical data are expressed as the mean ± SEM. n = 4 in *db/m* group and n = 9-10 mice in *db/db*, *db/db*-GKO group (**a**). n = 9-10 mice (**b-c**). *p < 0.05, **p < 0.01.

5. The authors state that “*db/db* GKO mice had a feathered appearance”. What does it mean?

R: This was a writing error. We apologize for that. We have changed the phrase to “*db/db* GKO mice featured improved first-phase insulin and C-peptide secretion starting at 9 weeks of age (Fig. 3q and Extended Data Fig. 3l) compared to *db/db* mice.”

C: OK.

6. The authors report that, in *db/db* mice, hyperinsulinemia is noted by 10 days of age and frank hyperglycemia with glucose values of 10-15 mM by 8 wk of age. The age dependent increase in plasma Gal3 (Fig. 3, panel 1) seems more a consequence than a contributing factor to diabetes development. Please discuss this issue.

R: In *db/db* mice, hyperglycemia is usually noted by 8 weeks of age. The plasma Gal3 levels increased significantly in *db/db* mice compared with *db/m* mice at 7 weeks of age (Fig. 1b in the original manuscript).

Fig. 1b in the original manuscript. Gal3 levels in the plasma of *db/m* and *db/db* mice (7 to 14 weeks old; n=3 mice). Two-tailed Student's t test was used, and statistical data are expressed as the mean ± SEM. *p < 0.05; **p < 0.01.

C: Knowing that hyperglycemia typically occurs approximately 8 weeks after the onset of hyperinsulinemia in this strain, and that hyperglycemia is a result of β cells failing to produce sufficient insulin to counteract insulin resistance, it is reasonable to infer that the functional decline of beta cells by 7 weeks of age is already significantly advanced. Therefore, while the relationship between blood glucose and plasma Gal3 levels indicates an association and does not establish causality or its direction, considering the pathophysiology of type 2 diabetes, these data suggest that

the increase in Gal3 likely represents a consequence rather than a contributing factor to the glucose metabolism derangement. In addition, the discovery that the blood glucose levels of *db/db* GKO mice plateaued from 9 weeks onward (refer to Figure 10 in the rebuttal) eliminates the possibility of Gal3 playing a causal role in promoting the development of diabetes.

R: At 7 weeks old, the blood Gal3 levels already showed 87.5% increase than the control mice. It's also reasonable to infer that Gal3 levels could be high before 7 weeks old. Most importantly, If Gal3 is a consequence rather than a cause, then the Gal3 KO mice or Gal3 Lyszre mice, or Gal3 inhibitors/antibody treatments (Fig. 3, 4 and 7 in the 2nd revised manuscript) won't give the phenotype. Again, if Gal3 is just a consequence, in MIN6 cells and primary islets (Fig. 2 in the 2nd revised manuscript), the addition of exogenous Gal3 protein to the medium won't give the phenotype. Thus, the data from both *in vivo* and *in vitro* experiments demonstrated that Gal3 is a contributing factor rather than a consequence.

We don't understand how the reviewer drew the conclusion that "the discovery that the blood glucose levels of *db/db* GKO mice plateaued from 9 weeks onward (refer to Figure 10 in the rebuttal) eliminates the possibility of Gal3 playing a causal role in promoting the development of diabetes.". In this model, the Gal3 KO ameliorated the increase of blood glucose in *db/db* mice. In fact, these results support the conclusion that Gal3 led to β -cell dysfunction. It's accepted that several factors besides Gal3 contribute to the development of β -cell dysfunction and diabetes in pathological conditions. The blood glucose is the net result of different factors *in vivo*. It's reasonable to see the phenotype after 9 weeks old since Gal3 is not the only contributing factor *in vivo*.

7. The authors report apparently contrasting or unclear data regarding age and blood glucose levels of *db/db* mice. It is not clear what is the age of *db/db* mice in panel (n) of figure 3 and in panel (r) of Fig.3. Basal blood glucose levels (time 0) of *db/db* mice are much higher in panel (n) than in graphs of panel (o). Yet, at 9, 12, and 14 wks the *db/db* mice should have already developed frank diabetes.

R: As Fig. 3k (in the original manuscript) mentioned, in Fig. 3n and Fig. 3r (in the original manuscript), the ages of the *db/db* mice were 15 weeks and 16 weeks, respectively. As shown in Fig. 10, the basal blood glucose levels (time 0) of *db/db* mice gradually increased as the mice aged, with approximately 200 mg/dl at 9 weeks old and approximately 300 mg/dl at 15 weeks old. However, the blood glucose of *db/db* GKO mice stopped increasing from 9 weeks of age, and the 2 groups showed significant differences at 15 weeks of age. The ITT experiment (Fig. 3n in the original manuscript) in *db/db* mice was performed at 15 weeks old, so the basal blood glucose levels were much higher in Fig. 3n (in the original manuscript) than in the graphs of Fig. 3o (at 9,12, and 14 weeks, in the original manuscript).

Fig. 10 Fasting blood glucose of *db/db* and *db/db* GKO mice. n= 5-7 mice (male, 8- 15 weeks old) ***P<0.001.

C: It is possible that the observed disparity is attributable to the utilization of distinct unit scales. However, it appears to me that there is a significant difference in the baseline blood glucose levels of *db/db* mice between panel n and panel o at the 14-week mark.

R: We used the same unit scales and replotted Fig 3o (in the original manuscript) and Fig. 3n (in the original manuscript). The basal glucose levels between 14 weeks and 15 weeks in *db/db* mice are comparable (Fig. 21).

Fig. 21 a-b, IPGTT (a) and ITT (b) in *db/db* and *db/db* GKO mice. n=5-7 mice. Two-tailed Student's t test was used, and statistical data were expressed as the mean \pm SEM. *p < 0.05; **p < 0.01; * p < 0.001.**

8. The authors present unclear data on insulin secretion, as in Fig 3, panel (p) and Extended Data Fig. 3, panel (l), it seems that first phase insulin secretion is impaired in both WT and GKO mice starting from 12 wk.

R: In Fig. 3p and Extended Data Fig. 3l, the first phase insulin secretion is impaired in *db/db* GKO mice starting from 12 weeks, but in *db/db* mice, it is from 9 weeks.

C: The authors are correct; we confused the animal experimental models (*db/db*, not HFD).

9. The authors do not provide convincing proof that Gal-3 from infiltrated macrophages plays a role in acute macrophage induced β -cell dysfunction. Insulin levels (production) are similar in islets derived from normal chow (NC) or *db/db* mice treated with vehicle and exposed to high glucose conditions (Fig 2, panels c and d). This does not support the conclusion that Gal3 from infiltrated macrophages impairs GSIS. Only exogenous Gal3 reduces GSIS in islets from both NC and *db/db* mice, although to a greater extent in islets from *db/db* mice than in islets from NC mice. This points in favor of some other factor responsible for the greater reduction of GSIS in the islets of *db/db* mice. The finding that Gal3 from macrophages of WT-NC mice does not affect GSIS in MIN6 cells (extended data Fig.2, panels I and J) supports this conclusion.

R: The GSIS experiment reflects the ability of β cells to secrete insulin when exposed to high- glucose conditions after low-glucose conditions. The insulin fold change instead of absolute insulin levels indicates the β -cell function ability. Many papers from human and animal studies have shown that during the early stage of diabetes^{11, 36-38}, pancreatic β cells tend to secrete more insulin in the basal state but show less response to high-glucose stimulation (change in insulin). Some papers even report GSIS results as the insulin fold change instead of insulin levels³⁹⁻⁴¹. Consistently, the islets from *db/db* mice had higher basal insulin levels compared to those of NC-fed mice (0.021 vs. 0.008 ng/ μ g Pro). Upon high-glucose stimulation, however, the insulin fold change increased approximately 20 times in islets derived from NC mice (Fig. 11a, blue bar), while in islets from *db/db* mice, the insulin levels increased only approximately 7 times, as shown in Fig. 11b (blue bar). Thus, our data support the

conclusion that Gal3 from infiltrated macrophages impairs GSIS.

Fig. 11 GSIS with Gal3 treatment (250 ng/ml) for 6 h in primary islets. (a) NC mice, (b) *db/db* mice. Two-tailed Student's t test was used, and statistical data are expressed as the mean \pm SEM. n = 4-6 mice (male, NC: 12 weeks old, *db/db*: 17 weeks old). *p < 0.05; **p < 0.01.

C: This is an elegant argument to support a role for macrophage-derived Gal3 in GSIS impairment

despite the lack of differences in GSIS between islets derived from *db/db* and those derived from WT NC mice. Physiopathologically speaking, however, what matters is the absolute quantity of insulin produced following stimulation with glucose, which is the same between *db/db* and WT NC mice. The data rather suggest that Gal3 from infiltrated macrophages may increase basal insulin levels, presenting a completely different meaning and potential metabolic consequences. However, the fact that the addition of exogenous Gal3 in basal conditions does not modify insulin production in the islets of *db/db* (and also NC, see Fig. 2 panels d and c) does not support this explanation and argues in favor of the effect of some other inflammatory factor, aside from Gal3, on these changes. In short, the effect of macrophages on insulin production does not seem to be dependent on Gal3.

The effect of macrophages on basal insulin production is also observed in MIN6 cells, but in the opposite way compared to what is seen in the islets. In fact, both WT and KO macrophages induce a reduction in insulin levels."

R: We used the same unit scales and replotted Fig 2d and 2c (in the 1st revised manuscript), and found that the insulin secretion (the absolute quantity) in the pancreatic islets of *db/db* mice under high glucose stimulation was significantly lower than that in the islets of NC mice (Fig. 22).

Additionally, we generated the macrophage/neutrophil Gal3-specific KO mice and found that these mice improved GTT and GSIS, further supporting the role of Gal3 in macrophages on β -cell function (see response to Reviewer #2, Fig. 18b-h).

Fig. 22 GSIS with Gal3 treatment (250 ng/ml) for 6 h in primary islets. a, NC mice (Fig. 2c in manuscript). b, *db/db* mice (Fig. 2d in manuscript). c, Insulin secretion of high glucose (16.8 mM) stimulated vehicle group in (a) and (b). Two-tailed Student's t test was used, and statistical data are expressed as the mean \pm SEM. n = 4-6 mice (male, NC: 12 weeks old, *db/db*: 17 weeks old). *p < 0.05; **p < 0.01.

R: With regard to the results in macrophages from NC-fed mice and MIN6 cell coculture experiments, in the NC-fed state, macrophages are not inflamed and secrete less Gal3^{24,42,43}. This would explain why macrophages from NC-fed mice did not affect β -cell function. This is consistent with the *in vivo* finding that the function of islets from NC-fed mice was not reduced.

C: Gal3 levels in the conditioned medium from the co-culture system of MIN6 cells are much higher in WT-NC macrophages than in KO-NC macrophages (or the absence of macrophages, as shown in extended data Fig. 2, panel I). These data support a role for macrophages independent of Gal3 production. The statement that 'This is consistent with the *in vivo* finding that the function of islets from NC-fed mice was not reduced' does not make sense. It is unclear what 'not reduced' refers to, as we are discussing WT mice on a standard diet (i.e., controls)

R: In the co-culture system, the Gal3 level in the WT-NC macrophage group is higher than in the KO group. However, the levels are still very low (approximately 1.5 ng/ml). Moreover, we found that in physiological conditions, there was no endogenous Gal3/CACNG1 interaction (**Fig. 24**). However, in pathological conditions, Gal3 levels are increased and there was endogenous Gal3/CACNG1 interaction (**Fig. 6b** in the revised manuscript), demonstrating that in physiological conditions (normal chow-fed mice), although there are certain levels of Gal3, it's not high enough to bind to CACNG1 and impair β -cell function. When we mentioned that "This is consistent with the *in vivo* finding that the function of islets from NC-fed mice was not reduced", we meant that this is consistent with the *in vivo* finding that the function of islets from WT and KO mice on NC was not different (**Extended Data Fig. 3f** in the original manuscript).

R: Additionally, we depleted macrophages in HFD-fed mice by clodronate treatment and found that there was a marked decrease in Gal3 levels in the circulation, islets and supernatant of islets (Fig. 12d, e and f) in the clodronate treatment group, along with decent macrophage depletion in the islets (Fig. 12g). Importantly, clodronate treatment led to an increase in GSIS in primary islets, glucose tolerance and insulin secretion in HFD-fed mice, also supporting the idea that Gal3 from infiltrated macrophages impairs GSIS (see response to Question #1 of Reviewer #3).

C: Together with the previous data, this new set of data demonstrates that macrophages play a role in insulin resistance and the development of type 2 diabetes. A reduction in Gal3 levels is associated with these favorable metabolic effects. These new experiments confirm the association and, in light of previous studies, particularly those conducted on isolated islets and MIN6 cells, suggest that the effects of macrophages are independent of their production of Gal3 to some extent (see my comment above to this issue).

R: As for this question, we generated macrophage Gal3 KO mice using Cre-Loxp system and found that these mice on HFD showed improved GTT, along with higher insulin secretion in primary islets from the macrophage Gal3 KO mice compared to controls. Thus, these results suggested that the Gal3 in the macrophages plays a role in β -cell dysfunction. Please see the response to Reviewer #2 (**Fig. 18b-h**).

The authors do not provide convincing proof that "KO (or pharmacological inhibition) of Gal3 was accompanied by a significant increase in β -cell mass". In fact, data demonstrate that the percentage of the islet area (fractional area) occupied by β -cells is increased, not the mean β -cell area. As the size of the islets is greatly reduced in the KO (and pharmacological treated) mice (see panels a-f of Fig 4, and panels a-c of Extended Data Fig. 6), the mean islet area must be measured, and the mean β -cell area calculated by multiplying the fractional β -cell area by the mean islet area and dividing by 100.

R: We thank the reviewer for the comments on the statement about β -cell mass. Actually, the data we showed were the β -cell area expressed as the percentage of islets. We measured the area of β cells and the islet where they are located, and then the percentage of the islet area occupied by β cells was calculated by using the β -cell area dividing by the islet area and multiplying by 100. We have corrected " β cell mass" to " β cell area (% islet)".

C: I apologize if the request was unclear. What the Authors are demonstrating is an increase in the average fractional area of β cells (i.e., the percentage of the islet occupied by beta cells). The islets of KO mice are smaller, so the data presented does not offer information on β cell mass per islet. It is possible that the β cell mass per islet could even be reduced in KOs if the percentage decrease in average islet size is greater than the fractional increase in β cells. Therefore, the data shown do not provide useful information regarding changes in β cell mass.

R: As suggested, we reanalyzed the β -cell mass. The β -cell mass was calculated by dividing the insulin-positive area by the total pancreatic area and multiplied by the pancreatic weight. The results showed that the β -cell mass was comparable between WT and Gal3^{+/-} mice on HFD and between *db/db* and *db/db* GKO mice (Fig. 23a and b).

Fig. 23 β -cell mass in WT and Gal3^{+/-} on HFD-fed mice (a), *db/db* and *db/db* GKO mice (b). Two-tailed Student's t test was used, and statistical data are expressed as the mean \pm SEM. n= 4 (a), n=8 (b). HFD: 16 weeks from 8 weeks of age. *db/db*:16 weeks of age.

10. The authors show protein-protein interaction between Gal3 and CANG1, which represents the main novelty of the study. However, this issue deserves to be investigated in more physiological conditions than those of MIN6 cells transfected with CANG1 cmc plasmid in the presence of added GFP-tagged Gal3. As only exogenous Gal3 (i.e., added Gal3), but not endogenous Gal3, reduces GSIS in mice islets and MIN6 cells (Fig 2, panels c and d and extended data Fig.2, see panels I and J), coimmunoprecipitation of Gal3 and CACNG1 should be assessed in mice islets from WT and KO *db/db* mice and in MIN6 cells cocultured with intraperitoneal macrophages from Gal3 KO and WT mice. This to exclude that what is observed in experiments already performed is an artifact due to the addition of exogenous Gal3, as it actually happens for the reduction of GSIS.

R: We thank the reviewer for the comment “The authors show protein–protein interaction between Gal3 and CANG1, which represents the main novelty of the study.” We agree with the reviewer that this issue should be investigated fully. First, we showed endogenous Gal3 and CACNG1 colocalization by Co-IP in the MIN6 and macrophage coculture system and immunofluorescence staining in islets from *db/db* and HFD-fed mice using Gal3 and CACNG1 antibodies (please see Fig. 6c and d in response to Question #7 of Reviewer #1). Second, we proved that the interaction of Gal3 and CACNG1 is direct by the PLA assay and BLI assay (please see Figure 6e and f in response to Question #7 of Reviewer #1). Third, we found the binding sites in CACNG1 and proved that the binding occurs through glycosylation (please see Fig. 6a and b in response to Question #7 of Reviewer #1). Finally, we generated CACNG1 KO and β - cell-specific CACNG1 KO mice and showed that CACNG1 is critical for β -cell function and in vivo glucose tolerance (please see Fig. 3 in response to Question #2 of Reviewer #1 and Fig. 13d-g in response to Question #2 of Reviewer #3).

C: These new experiments demonstrate that CANG1 is critical for β cell function and glucose tolerance. Gal3 and CANG1 colocalize in islets from *db/db* and HFD-fed mice. One of the points to clarify, as originally requested by this Reviewer, is whether the interaction occurs physiologically or only in obese/diabetic animals; that is, does the interaction have pathophysiological relevance?

Unfortunately, the new experiments did not include metabolically healthy WT mice. Gal3 functions as an endocytic adapter that co-clusters glycosylated cargos and membrane glycosphingolipids,

leading to both membrane bending and clathrin-independent formation of the endocytic pit (PMID: 24837829). In addition, the formation of a cross-linked lattice between galectin-3 and cell surface glycoprotein receptors was shown to inhibit endocytosis of some receptors and regulate their availability for signaling. Therefore, it is not surprising that Gal3 and CACNG1 colocalize in islets from mice, including *db/db* and HFD-fed mice, and pathomechanistic conclusions cannot be drawn from these findings.

R: As suggested, we performed the immunofluorescence staining in islets from *db/m* and NC-fed mice using Gal3 and CACNG1 antibodies, and the results showed that there was no endogenous binding of Gal3 and CACNG1 in islets either from *db/m* or from NC fed mice (**Fig. 24**). Suggesting that the interaction is not a physiological but a pathological event.

Fig. 24 Colocalization of Gal3 and CACNG1 was detected using antibodies against the endogenous proteins in islets from *db/m* and NC mice. NC: 16 weeks old, C57BL6J, male; *db/m*: 17 weeks old, BKS, male).

REVIEWERS' COMMENTS

Reviewer #2 (Remarks to the Author):

The authors have addressed all my concerns with the elegant experiments.

Reviewer #3 (Remarks to the Author):

The Authors addressed all the issues raised, eventually by providing data from new experiments. Overall the manuscript has improved. However, the Authors' responses are convincing only for some issues. Therefore, several differences in the interpretation of data remain, but I believe that they cannot be further reduced.